# Defending Multimodal Backdoored Models by Repulsive Visual Prompt Tuning

**Zhifang Zhang**[1,2]  **Shuo He**[3*]  **Haobo Wang**[4]  **Bingquan Shen**[5]  **Lei Feng**[1*]

[1]Southeast University   [2]University of Queensland   [3]Nanyang Technological University

[4]Zhejiang University   [5]National University of Singapore

## Abstract

Multimodal contrastive learning models (e.g., CLIP) can learn high-quality representations from large-scale image-text datasets, while they exhibit significant vulnerabilities to backdoor attacks, raising serious safety concerns. In this paper, we reveal that CLIP's vulnerabilities primarily stem from its tendency to encode features beyond in-dataset predictive patterns, compromising its visual feature resistivity to input perturbations. This makes its encoded features highly susceptible to being reshaped by backdoor triggers. To address this challenge, we propose Repulsive Visual Prompt Tuning (RVPT), a novel defense approach that employs deep visual prompt tuning with a specially designed feature-repelling loss. Specifically, RVPT adversarially repels the encoded features from deeper layers while optimizing the standard cross-entropy loss, ensuring that only predictive features in downstream tasks are encoded, thereby enhancing CLIP's visual feature resistivity against input perturbations and mitigating its susceptibility to backdoor attacks. Unlike existing multimodal backdoor defense methods that typically require the availability of poisoned data or involve fine-tuning the entire model, RVPT leverages few-shot downstream clean samples and only tunes a small number of parameters. Empirical results demonstrate that RVPT tunes only 0.27% of the parameters in CLIP, yet it significantly outperforms state-of-the-art defense methods, reducing the attack success rate from 89.70% to 2.76% against the most advanced multimodal attacks on ImageNet and effectively generalizes its defensive capabilities across multiple datasets. The code is publicly available in our GitHub repository: https://github.com/zhangzf01/RVPT.

## 1   Introduction

Recently, Contrastive Language-Image Pretraining (CLIP) [52] has emerged as a powerful base model across multiple domains. Unlike traditional models that rely on uni-modal supervision, CLIP aligns visual and textual signals to learn rich representations from numerous image-text pairs in the pre-training stage. This unique training paradigm enables its impressive performance in open-vocabulary visual recognition [64] and robustness to distribution shift [13].

However, despite CLIP's success in visual recognition, recent research [5, 4] has revealed its vulnerabilities against backdoor attacks in downstream tasks. The vulnerabilities can be roughly categorized into two aspects. First, concerns arise from its training data collection process: the data is noisy, uncurated, and sourced from the web, which facilitates adversarial poisoning of the image-text pairs. Second, CLIP demonstrates heightened susceptibility to a smaller number of poisoned training samples: recent studies [5] have empirically shown that poisoning CLIP may require injecting orders of magnitude fewer poisoned data into the training dataset than poisoning conventional supervised

---

*Corresponding author.

39th Conference on Neural Information Processing Systems (NeurIPS 2025).

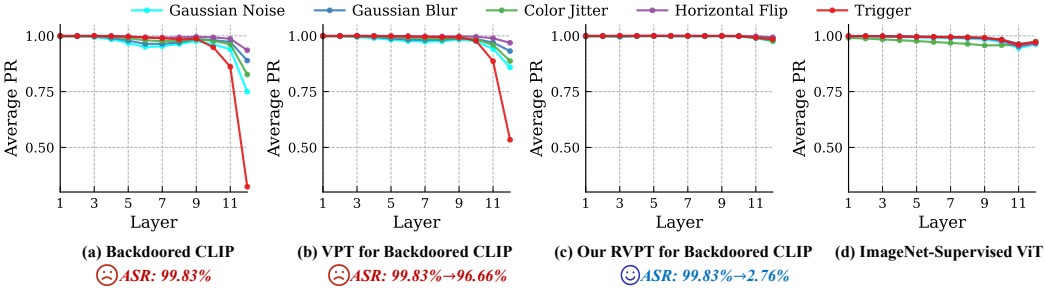

Figure 1: **Perturbation Resistivity** (PR) across different layers of the encoders under various perturbations, including the trigger pattern. A higher PR value indicates less sensitivity to input perturbation. Four encoder variants are evaluated on ImageNet: (a) ViT in backdoored CLIP [52]. (b) ViT in backdoored CLIP tuned with Visual Prompt Tuning (VPT) on ImageNet. (c) ViT of the backdoored CLIP tuned with our RVPT on ImageNet. (d) ViT trained exclusively on clean ImageNet [51]. Specifically, CLIP is backdoored by the trigger of BadCLIP [35]. Detailed experimental settings and additional PR results for CLIP backdoored by other triggers can be found in Appendix B.

models. Once poisoned during pre-training, the backdoored CLIP will inevitably classify images with an adversarially designed trigger into the predefined target class.

CLIP's vulnerabilities to backdoor attacks have inspired the development of various defense methods. However, most of these methods necessitate fine-tuning the entire model's parameters [3] or require the availability of poisoned data [68, 24, 67], which could be resource-intensive or even unrealistic. Thus, to achieve an efficient, effective, and practical defense, we are dedicated to developing a defense method based on visual prompt tuning (VPT) [26], utilizing only a small portion of downstream clean samples to encourage the model to ignore the trigger feature in the poisoned images. [1] However, simply adopting VPT proves ineffective experimentally (see Figure 1(b)), as the trigger feature cannot be extracted from clean samples, preventing VPT from learning how to suppress them.

In this paper, we find that CLIP's visual embeddings remain highly sensitive to small input perturbations even after VPT, leaving it susceptible to potential backdoor attacks. To quantify this sensitivity, we introduce the *Perturbation Resistivity* (PR) metric, defined as the cosine similarity between the visual embedding $f^v(\boldsymbol{x})$ of an image $\boldsymbol{x}$ and that of its perturbed counterpart: $\boldsymbol{x} + \boldsymbol{\delta}$:

$$\text{PR}(\boldsymbol{x}, \boldsymbol{\delta}) = \cos(f^v(\boldsymbol{x}), f^v(\boldsymbol{x} + \boldsymbol{\delta})), \tag{1}$$

where $f^v(\cdot)$ denotes the visual feature extractor. A lower PR value implies larger visual embedding drift due to input perturbations, hence greater sensitivity and higher susceptibility to backdoor triggers. To illustrate this vulnerability of CLIP, we apply a suite of perturbations to ImageNet samples and plot the layer-wise mean PR for several model variants. The results in Figure 1 show that the deeper layers of standard CLIP (Figure 1(a)) and standard CLIP tuned with VPT (Figure 1(b)) exhibit significantly lower PR than a ViT trained *exclusively* on clean ImageNet (Figure 1(d)), confirming that CLIP's embeddings are easily distorted. Because a backdoor trigger acts as a structured perturbation, this low PR allows the trigger to maliciously reshape the encoded visual features layer by layer, enabling a successful attack. We therefore regard PR as a proxy for backdoor robustness, which means the *backdoor-robust* CLIP should have a higher PR than *backdoor-vulnerable* CLIP (i.e., the standard CLIP). Our analysis suggests that CLIP's low PR originates from its large-scale, noisy pre-training dataset: the encoder captures many off-dataset, non-predictive features, which magnify the impact of perturbations. In contrast, the ImageNet-supervised ViT only learns in-dataset predictive features of ImageNet, prohibiting it from encoding irrelevant features beyond ImageNet and achieving a much higher PR. Hence, an effective defense (ours shown in Figure 1(c)) should suppress the irrelevant features CLIP inherently captures and guide it to focus on in-dataset predictive patterns, thereby enhancing both its perturbation resistivity and backdoor robustness.

Inspired by this finding, we propose a novel approach termed Repulsive Visual Prompt Tuning (RVPT), which improves CLIP's PR by making it exclusively encode the predictive features in the downstream tasks in a few-shot manner. As illustrated in Figure 2, RVPT jointly optimizes two

---

[1]We include the detailed reasons for using VPT to defend backdoored CLIP in Appendix A.

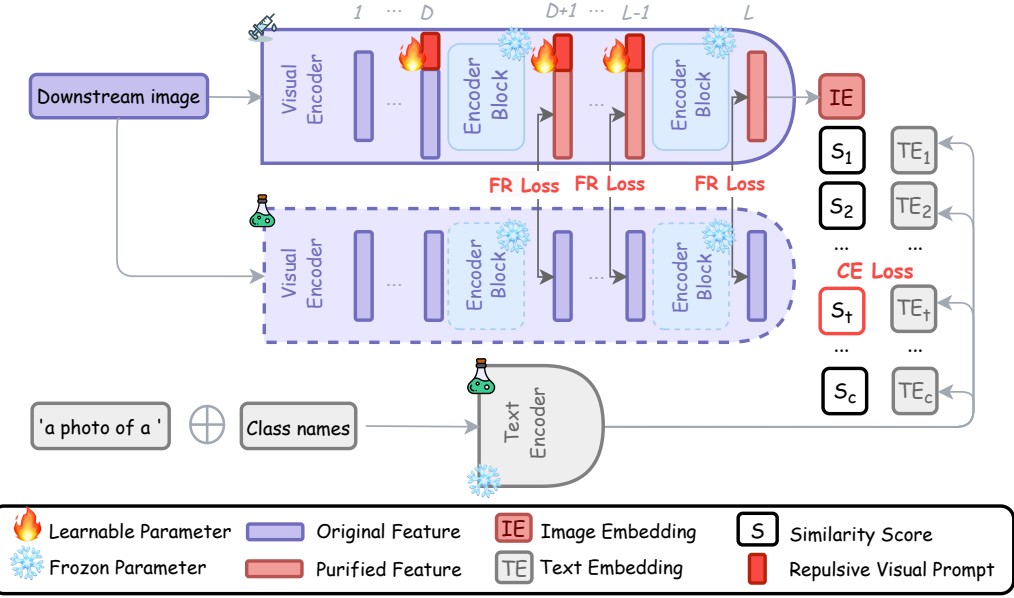

Figure 2: **Illustration of RVPT.** RVPT concatenates the features from later layers with tunable visual prompts while keeping the original parameters of CLIP frozen. To optimize the visual prompts, RVPT employs both cross-entropy (CE) loss and feature-repelling (FR) loss. The FR loss minimizes the mean cosine similarity between the prompted features and the original features across layers. Meanwhile, the CE loss ensures the clean accuracy of the prompted model. Together, these losses guide the model to encode only in-dataset predictive features that contribute to CE loss optimization, thereby enhancing its backdoor robustness.

objectives: (1) a multi-layer feature-repelling loss that minimizes the cosine similarity between prompted features and original CLIP features at each layer, and (2) cross-entropy loss for downstream task accuracy. This mechanism forces the model to only extract features that contribute to CE loss optimization during training, while ignoring non-essential features (e.g., noise or trigger pattern). As a result, Figure 1(c) shows that RVPT significantly improves the PR of CLIP on ImageNet, hence effectively defending against the state-of-the-art multimodal backdoor attack. Moreover, extensive experiments demonstrate our method's effectiveness across multiple datasets and backdoor attacks.

## 2   Related Work

**Backdoor attacks and defenses on supervised learning.** Backdoor attacks have become an increasingly serious security issue because more and more practitioners choose to adopt third-party datasets, platforms, or even backbones to reduce costs. Research on backdoor attacks has primarily focused on designing triggers that enhance stealthiness [7, 57] and effectiveness [39, 46]. For backdoor defenses, researchers employ various strategies to mitigate backdoor attacks, including: (1) pre-processing defense [56] (2) pre-training defense [6] (3) post-training defense [76, 62] (4) test-time defense [15]. Specifically, our work aligns with the post-training defense category, which assumes that defenders do not have access to the poisoned training data and must eliminate the backdoor threat that has already been implanted into the model. There are two main strategies for post-training defense. (1) Pruning-based Methods [38, 63]: These approaches focus on identifying and removing the most suspicious neurons. (2) Fine-tuning-based Methods [60, 76, 45, 32]: These methods purify the backdoored model by directly tuning its parameters with clean data.

**Backdoor attacks and defenses on vision-language models.** Although vision-language models (VLMs) [52, 25, 37] have demonstrated significant improvements across various fields, recent research [5, 69] has revealed vulnerabilities of contrastively pre-trained VLMs to backdoor attacks, drawing substantial attention in the field of backdoor attacks and defenses on CLIP-like models. In

terms of attacks, BadCLIP [35] optimizes visual trigger patterns within a dual-embedding guided framework, rendering the attack effective and undetectable. Moreover, many backdoor attack methods on multimodal large language models [41, 42, 34, 48, 43, 33, 70, 40] have been proposed recently. Defense strategies can be primarily categorized into two categories: pre-training defense and post-training defense. For pre-training defense [68, 24, 67], SAFECLIP [68] prevents learning from poisoned examples by dynamically dividing the training data into the safe and risky sets by cosine similarity and only applying uni-modal contrastive loss to different modalities of the risky set. For fine-tuning defense [3, 66, 27], CleanCLIP [3] reduces the impact of spurious relationships of backdoor attacks by enforcing the model to realign representations for independent modalities. There are also some work focusing on test-time defense on multimodal backdoor attacks [49, 18, 21].

**Prompt learning for vision-language models.** Lately, various fine-tuning methods have been introduced to further enhance the impressive capabilities of vision-language models (VLMs) across diverse downstream tasks [29, 22, 75, 65], with prompt learning [74, 53, 72] emerging as a particularly efficient and effective approach. Specifically, CoOp [74] first introduces prompt learning to VLMs and proposes a method that optimizes the context of the text prompt, e.g., *"a photo of a <CLS>."* while keeping the class name tokens fixed. VPT [26] adjusts the visual features by concatenating the image or image features with the learnable prompt. Besides, BadCLIP [1] injects backdoor during the prompt learning stage, enabling the malicious alteration of text features through trigger-aware prompts for a powerful attack. We notice that despite the effectiveness of prompt learning for enhancing the model's robustness [27, 30, 8, 73, 11], limited research has explored its use in defending against multimodal backdoor attacks. In particular, current post-training backdoor defense methods for VLMs mainly focus on fine-tuning the whole backdoored models, overlooking the potential of prompt learning. To address the gap, we propose RVPT, which incorporates learnable parameters into features to defend against multimodal backdoor attacks while freezing the model's parameters.

## 3 The Proposed Approach

**Overview.** In this section, we first outline the basic settings of the threat model and defender's goal in Section 3.1, followed by a detailed illustration of how the adversary launches backdoor attacks on CLIP in Section 3.2. Eventually, in Section 3.3, we present Repulsive Visual Prompt Tuning, designed to defend backdoored CLIP against multiple multimodal backdoor attacks.

### 3.1 Basic Settings

**Threat model.** We assume the adversary has access to the training dataset and can produce a backdoored model by injecting crafted poisoned data into the original dataset. For multimodal attacks, the poisoned data usually consists of images with the trigger pattern and the proxy captions for the target class. Once trained on the poisoned dataset, the model will misclassify inputs with the trigger pattern into the predefined target class while maintaining classify correctly for the benign inputs.

**Defender's goal.** We suppose that the defender has access to the backdoored model and a limited set of clean downstream data. The defender's goal is to obtain a new model utilizing both the backdoored model and the available clean data, wherein the backdoor effects are mitigated while preserving the benign performance of the new model on the downstream clean data. Furthermore, the defender has full control over the recovering process of the backdoored model.

**Clarification and practical motivation.** Although the above setting assumes the defender lacks access to the original pre-training data, this assumption is both realistic and practically important. First, the backdoor robustness of many multimodal models depends on CLIP, whose pre-training data is *entirely unavailable* to the public or defenders. Even though some models partially disclose data sources (e.g., LAION, CC3M), most real-world multimodal models are trained on large-scale, proprietary, or mixed Internet data, making it infeasible for defenders to reconstruct the same distribution. Recent studies also show that backdoored CLIP models can transfer their vulnerabilities to other multimodal models where CLIP serves as the visual encoder, highlighting the need to defend data-closed models. Second, even when pre-training data is partially available, leveraging it for defense remains highly challenging. Because multimodal datasets are vast and Internet-sourced, locating and removing poisoned samples is computationally prohibitive. Using a surrogate dataset for fine-tuning (e.g., CC3M) is an alternative, but it causes feature mismatches and significantly

degrades defense performance, as verified experimentally. In contrast, our setting targets defenders with only a small amount of trusted downstream data—a realistic and widely encountered scenario. Such a *data-limited but model-accessible* defense setting frequently appears in practice. For example, enterprises or institutions may adapt pre-trained CLIP models to downstream tasks (e.g., medical imaging, rare-disease recognition, or industrial inspection) but only have few-shot data due to privacy or scarcity. In these cases, defenders can access the backdoored model but not its large-scale training data. Our RVPT method is designed for this setting: it provides an end-to-end defense that adapts CLIP to downstream tasks while mitigating backdoors using only a few clean samples.

## 3.2 Backdoor Attacks on CLIP

In general, the CLIP model comprises a visual encoder $\mathcal{V}(\cdot)$, a textual encoder $\mathcal{T}(\cdot)$, and projection matrices that project the encoded representations into a joint feature space: $\boldsymbol{P}_\mathrm{I}$ for the image modality and $\boldsymbol{P}_\mathrm{T}$ for the text modality. The training dataset contains $N_1$ image-text pairs represented as $\mathcal{D} = \{(\boldsymbol{x}_i, \boldsymbol{t}_i)\}_{i=1}^{N_1}$, where $\boldsymbol{t}_i$ serves as a short caption for the corresponding image $\boldsymbol{x}_i$.

During the backdoor attack on CLIP in the training stage, adversaries typically inject a small number of $N_2$ poisoned image-text pairs denoted as $\mathcal{D}_p = \{(\boldsymbol{x}_i^p, \boldsymbol{t}_i^p)\}_{i=1}^{N_2}$. $\boldsymbol{x}_i^p$ represents a poisoned image obtained by modifying the original image $\boldsymbol{x}_i$ with the trigger pattern $\Theta$, while $t_i^p$ denotes the proxy caption for the target class $y_t$. Consequently, the original benign training dataset $\mathcal{D}$ can be poisoned, resulting in: $\widetilde{\mathcal{D}} = \{\mathcal{D} \cup \mathcal{D}_p\}$. During training, CLIP is optimized on $\widetilde{\mathcal{D}}$ to maximize the cosine similarity of positive pairs, expressed as $\phi(\widetilde{\boldsymbol{x}}_i, \widetilde{\boldsymbol{t}}_i) = \cos(\boldsymbol{P}_\mathrm{I}\mathcal{V}(\widetilde{\boldsymbol{x}}_i), \boldsymbol{P}_\mathrm{T}\mathcal{T}(\widetilde{\boldsymbol{t}}_i))$, and minimize the similarity of negative pairs $\phi(\widetilde{\boldsymbol{x}}_i, \widetilde{\boldsymbol{t}}_{j\neq i})$.

Upon training with $\widetilde{\mathcal{D}}$, the encoders of CLIP will be backdoored, denoted as $\{\widetilde{\mathcal{V}}(\cdot), \widetilde{\mathcal{T}}(\cdot)\}$. For the backdoored version of CLIP, the trigger pattern $\Theta$ will have a spurious correlation with the proxy caption of the target class $y_t$. Thus, during the inference stage, when the model encounters the images containing the trigger pattern, its prediction on the target class $y_t$ will increase significantly. In the meantime, the backdoored CLIP performs normally on benign inputs.

## 3.3 Repulsive Visual Prompt Tuning

In this part, we present Repulsive Visual Prompt Tuning (RVPT), a framework designed to defend against multimodal backdoor attacks. Our key insight is to make CLIP ignore trigger features of the attacked images by restraining CLIP's visual encoder to only encode in-dataset predictive features.

Formally, assume the defender is given the backdoored CLIP, denoted by $\{\widetilde{\mathcal{V}}(\cdot), \widetilde{\mathcal{T}}(\cdot)\}$ alongside a small set of clean labeled samples $\{\boldsymbol{x}_i, y_i\}_{i=1}^N$, $y_i \in \mathcal{Y} = \{1, \ldots, C\}$. In particular, our focus is on CLIP with the Vision Transformer [12] as its visual encoder. For clean input $\boldsymbol{x}_i$, the feature representation at the $d$-th layer of CLIP's visual encoder is $\boldsymbol{f}_i^d = [\boldsymbol{c}_i^d, \boldsymbol{e}_i^d]$, where $\boldsymbol{c}_i^d \in \mathbb{R}^{1 \times d_v}$ denotes the class embedding that will finally be projected to the vision-language joint space and $\boldsymbol{e}_i^d \in \mathbb{R}^{M \times d_v}$ are the patch embeddings. Specifically, $M$ and $d_v$ are hyperparameters representing the number of patches and dimensions of the visual embedding.

Assume the visual encoder has $L$ layers, then starting from the $D$-th layer, we introduce $L - D$ repulsive visual prompts, each of length $b$, defined as $\{\boldsymbol{p}^d \in \mathbb{R}^{b \times d_v}\}_{d=D}^{L-1}$. These prompts are concatenated with the features from each subsequent block recursively, following the formulation:

$$[\boldsymbol{c}_i^{d+1}, \boldsymbol{e}_i^{d+1}, \_] = \mathcal{V}^d([\boldsymbol{c}_i^d, \boldsymbol{e}_i^d, \boldsymbol{p}_i^d]), d = D, \ldots, L-1. \tag{2}$$

We then regard $\boldsymbol{c}_i^d, d \in [D+1, L]$ as the feature to repel and apply feature-repelling (FR) loss to minimize the cosine similarity between it and its counterpart $\boldsymbol{\sigma}_i^d$ in the original fixed backdoored visual encoder. For a batch of $N_b$ examples $\{\boldsymbol{x}_i, y_i\}_{i=1}^{N_b}$, the FR loss is computed as:

$$\mathcal{L}_{\mathrm{FR}} = \frac{1}{L-D} \sum_{i=1}^{N_b} \sum_{d=D+1}^{L} \cos(\boldsymbol{c}_i^d, \boldsymbol{\sigma}_i^d). \tag{3}$$

Additionally, we also optimize the prompts with the cross-entropy (CE) loss to avoid repelling the in-dataset predictive features and maintain clean accuracy, formulated as:

$$\mathcal{L}_{\mathrm{CE}} = -\sum_{i=1}^{N_b} \log\left(\frac{\exp(\widetilde{\phi}(\boldsymbol{x}_i, T(y_i))/\tau)}{\sum_{c=1}^{C} \exp(\widetilde{\phi}(\boldsymbol{x}_i, T(c))/\tau)}\right), \tag{4}$$

Table 1: **Main results.** We report ASR (↓ %), with CA (%) shown in parentheses on ImageNet1K.

| Method | BadNet | Blended | ISSBA | WaNet | TrojVQA | BadCLIP |
|---|---|---|---|---|---|---|
| No defense | 82.69 (63.04) | 98.52 (62.64) | 60.01 (61.72) | 87.18 (62.42) | 99.75 (62.81) | 99.83 (61.33) |
| CleanCLIP | 23.79 (57.91) | 0.25 (57.69) | 15.62 (59.20) | 11.10 (59.07) | 85.64 (58.22) | 89.70 (57.55) |
| Linear Probe | 3.05 (59.64) | 5.52 (59.69) | 0.08 (59.69) | 0.65 (59.66) | - | 99.70 (59.33) |
| **RVPT** | **0.05** (62.76) | **0.02** (62.36) | **0.01** (61.92) | **0.03** (62.48) | **0.11** (62.63) | **2.76** (61.81) |

where $\tau$ denotes a temperature parameter to control the logit distribution and is set as specified in [52], $T(\cdot)$ is the proxy caption of the input class, e.g., *"a photo of a <CLS>."* and $\widetilde{\phi}(\boldsymbol{x}_i, \boldsymbol{t}_j) = \cos(\boldsymbol{P}_{\mathrm{I}}\widetilde{\mathcal{V}}(\boldsymbol{x}_i), \boldsymbol{P}_{\mathrm{T}}\widetilde{\mathcal{T}}(\boldsymbol{t}_j))$ means cosine similarity of the outputs from the recovering encoders.

Thus, the overall loss for RVPT is defined as:

$$\mathcal{L} = \mathcal{L}_{\mathrm{CE}} + \alpha\mathcal{L}_{\mathrm{FR}}, \tag{5}$$

where $\alpha$ is the balancing factor controlling FR loss strength.

For further explanation, RVPT can be interpreted through the lens of natural selection [9]. To be specific, FR loss creates a challenging environment for visual prompt tuning to optimize CE loss. In order to simultaneously minimize CE loss and maximize the discrepancy between the prompted features and original features, the visual prompts must learn to select the most suitable features to optimize CE loss while ignoring those that are non-essential. Ultimately, this process leads to a more compact visual encoding mechanism, enhancing CLIP's perturbation resistivity to trigger patterns.

## 4 Experiments

### 4.1 Experiment Settings

**Models and benchmarks.** Following the settings of [3], to create a more realistic backdoored CLIP for backdoor defense, we fine-tune OpenAI's public checkpoint CLIP-400M [52] using 500K image-text pairs from the CC3M dataset [55], out of 1500 are poisoned with various types of backdoor attacks: BadNet [17], Blended [7], ISSBA [31], WaNet [47], TrojVQA [59] and BadCLIP [35]. We utilize ImageNet [10] to evaluate the performance of defense methods against all the aforementioned attacks, with the target class of the attacks set to banana. Additionally, we use Caltech101 [14] and OxfordPet [50] (a fine-grained dataset) to evaluate defense methods against BadNet, Blended, and WaNet. These attacks target "accordion" in Caltech101 and "samoyed" in OxfordPet, respectively. We compare RVPT with Linear Probe and CleanCLIP [3]. The implementation details of the backdoor attack methods, backdoor defense methods, and benchmarks will be illustrated in Appendix C.

**Evaluation criteria.** The evaluation metrics include Clean Accuracy (CA) and Attack Success Rate (ASR). CA is the model's accuracy on clean images, while ASR measures the proportion of poisoned images predicted as the target label. A lower ASR and decent CA stand for a successful defense.

**Implementation details.** Unless otherwise specified, we will use ViT/B32 [12] as CLIP's visual encoder. The prompt is initialized from a zero mean Gaussian distribution. We set the non-prompted depth $D = 3$, learnable context length $b = 50$, and balancing factor $\alpha = 2$. Moreover, the proxy caption of the class the same as the simple prompt engineering in the original paper of CLIP [52]: (1) *"a photo of <CLS>."* for ImageNet and Caltech101 (2) *"a photo of <CLS>, a type of pet."* for OxfordPets. For all datasets, we randomly sample 16 images per class while setting the batch size and the epoch number to 32 and 50. The loss is optimized with SGD with an initial learning rate of 0.002 decayed by the cosine annealing rule. We conduct experiments on eight NVIDIA RTX 3090 GPUs and the computational expenses are shown in Appendix D.

### 4.2 The Effectiveness of RVPT

In Table 1 and 2, we compare RVPT with CleanCLIP and Linear Probe against six attacks on ImageNet and three attacks on Caltech101 and OxfordPets. Before poisoning, CLIP's CA on ImageNet, Caltech101, and OxfordPets is 62.01%, 91.46%, and 86.42%, respectively.

Firstly, we can see *all attacks achieve a high ASR while maintaining decent CA with no defense*, indicating their successful executions. The CA loss of the poisoned CLIP

Table 2: **Main results.** We report ASR (↓ %), with CA (%) in parentheses on Caltech101, OxfordPets.

| Method | Caltech101 | | | OxfordPets | | |
| | BadNet | Blended | WaNet | BadNet | Blended | WaNet |
|---|---|---|---|---|---|---|
| No defense | 91.38 (93.06) | 92.69 (93.41) | 63.21 (92.86) | 86.83 (82.91) | 99.80 (85.10) | 87.97 (83.93) |
| CleanCLIP | 36.87 (91.18) | 1.14 (90.77) | 9.35 (91.54) | 25.72 (82.49) | 4.17 (83.41) | 12.61 (81.10) |
| Linear Probe | 1.22 (93.62) | 12.82 (93.41) | 1.04 (93.45) | 16.05 (77.74) | 2.63 (77.63) | 2.21 (77.71) |
| **RVPT** | **0.00** (94.02) | **0.00** (94.34) | **0.08** (93.89) | **0.30** (88.60) | **0.64** (88.87) | **1.59** (88.53) |

Table 4: **Results on emerging target class.** We report ASR (↓ %), with CA (%) shown in parentheses. (woTC) indicates RVPT is tuned in the dataset without the target class.

| Attack Type | ImageNet (woTC) | ImageNet | Caltech101 (woTC) | Caltech101 | OxfordPets (woTC) | OxfordPets |
|---|---|---|---|---|---|---|
| BadNet | 0.01 (62.50) | 0.05 (62.76) | 0.04 (93.71) | 0.00 (94.02) | 0.33 (86.84) | 0.30 (88.60) |
| Blended | 0.01 (62.63) | 0.02 (62.36) | 0.00 (93.89) | 0.00 (94.34) | 0.00 (87.19) | 0.64 (88.87) |
| WaNet | 0.00 (62.46) | 0.03 (62.48) | 0.00 (93.67) | 0.08 (93.89) | 0.00 (86.17) | 1.59 (88.53) |

Table 5: **Results on cross-dataset generalization.** We report ASR (↓ %), with CA (%) shown in parentheses. RVPT* means that the model is tuned on ImageNet1K using RVPT and tested on Caltech101 (target class: accordion) and OxfordPets (target class: samoyed).

| Method | Caltech101 | | | OxfordPets | | |
| | BadNet | Blended | WaNet | BadNet | Blended | WaNet |
|---|---|---|---|---|---|---|
| CleanCLIP | 36.87 (91.18) | 1.14 (90.77) | 9.35 (91.54) | 25.72 (82.49) | 4.17 (83.41) | 12.61 (81.10) |
| **RVPT*** | **2.32** (89.33) | **0.61** (89.91) | **1.80** (89.66) | **2.86** (82.65) | **0.53** (83.49) | **2.49** (82.66) |

varies across downstream datasets because to obtain a backdoored CLIP, fine-tuning on the poisoned CC3M dataset [55] changes CLIP's learned representations. These changes affect how CLIP performs on different datasets, accounting for varying levels of CA loss.

Secondly, although Clean-CLIP mitigates backdoor attacks in a zero-shot setting, it deteriorates CLIP's recognition performance and requires more computing resources. Moreover, Linear Probe performs well against the first four attacks. How-

Table 3: **Poisoned accuracy** (↑ %), defined as the model's classification accuracy on poisoned images.

| Model | BadNet | Blended | BadCLIP |
|---|---|---|---|
| *Clean Model* | *58.29* | *54.16* | *55.34* |
| Backdoored Model | 10.36 | 1.05 | 0.18 |
| CleanCLIP | 39.66 | 46.12 | 4.25 |
| Linear Probe | 28.02 | 14.94 | 5.64 |
| **RVPT** | **58.94** | **53.27** | **53.90** |

ever, it does not eliminate the trigger feature and thus performs the worst against BadCLIP, an attack specially designed for multimodal contrastive learning models. Since Linear Probe inevitably excludes CLIP's text encoder, it cannot be used to defend against TrojVQA, which targets the multimodal task of visual question answering. Finally, *RVPT effectively defends against all types of attacks while maintaining clean accuracy*. It demonstrates consistently strong defensive performance across all attacks and, notably, significantly reduces the ASR of BadCLIP [35], which is hard to defend by current defense methods. Moreover, as shown in Table 3, *RVPT's classification accuracy on poisoned images is also much higher than other defense method*s, highlighting its enhanced capability in maintaining correct predictions under backdoor attacks. Moreover, we observed that even a non-backdoored CLIP experiences a significant drop in PA, further supporting our claim of its low visual feature resistivity to input perturbation.

## 4.3 The Generalization of RVPT's Defensive Ability

We evaluate RVPT's generalization ability in defending backdoor attacks in three scenarios: (1) The target class of the attacks is absent from the tuning dataset but appears during testing. (2) The downstream dataset that will be attacked is different from the tuning dataset. (3) The downstream dataset that will be attacked shares different domains with the tuning dataset.

**Performance on the emerging target class.** We remove the samples belonging to the target class from the tuning dataset to evaluate RVPT's performance on the emerging target class, which measures its defense capability for unseen target classes in the dataset, and the results are shown in Table 4. Moreover, we also tested BadCLIP (woTC) for the target class of banana and the ASR (CA) is

Table 6: **Results on cross-domain generalization.** We report ASR (↓ %), with CA (%) shown in parentheses. We tune CLIP with RVPT on ImageNet1K and conduct CleanCLIP on CC3M.

| Dataset | Method | BadNet | Blended | BadCLIP |
|---|---|---|---|---|
| ImageNet-V2 | No defense | 86.55 (55.39) | 99.04 (54.83) | 99.89 (53.49) |
| | CleanCLIP | 31.38 (50.95) | 0.42 (50.96) | 92.04 (50.60) |
| | **RVPT** | **0.04** (53.85) | **0.02** (53.77) | **3.43** (52.53) |
| ImageNet-A | No defense | 92.97 (31.47) | 99.89 (31.22) | 99.97 (30.80) |
| | CleanCLIP | 59.11 (25.71) | 3.18 (27.10) | 98.18 (25.52)) |
| | **RVPT** | **1.64** (16.52) | **0.17** (16.84) | **12.84** (16.84) |
| ImageNet-R | No defense | 66.63 (67.11) | 97.94 (66.06) | 99.69 (65.49) |
| | CleanCLIP | 32.99 (61.81) | 2.54 (60.69) | 89.03 (60.93) |
| | **RVPT** | **0.76** (58.39) | **0.09** (58.46) | **6.75** (57.37) |
| ImageNet-S | No defense | 92.11 (41.73) | 97.16 (41.86) | 99.88 (40.19) |
| | CleanCLIP | 26.54 (34.62) | 0.26 (34.82) | 85.62 (34.44) |
| | **RVPT** | **0.02** (35.17) | **0.01** (35.34) | **1.67** (34.06) |

3.39 (61.89). Table 4 shows that for the backdoor attack of Blended and WaNet, RVPT performs even better in the absence of the target class, suggesting that RVPT has successfully filtered out the distinctive out-of-dataset pattern. Moreover, we can see that CA maintains except OxfordPets, which decreases by around 2%. The CA drop is due to this dataset containing only 37 classes, so the absence of 3% of the data can impact its accuracy. In summary, *RVPT's defensive capability effectively generalizes to attacks with emerging target classes that are not seen in the tuning dataset.*

**Performance on cross-dataset transfer.** To evaluate RVPT's defensive ability across datasets, we tuned CLIP with RVPT on ImageNet and tested it on Caltech101 and OxfordPets. The results are shown in Table 5. For all three attacks, although ASR increases slightly, it remains at a very low level. Additionally, RVPT's accuracy decreases and occasionally falls below that of CleanCLIP. We conjecture the reason is that ImageNet does not share some of the in-dataset predictive features with Caltech101 or OxfordPets. In some cases, The non-essential feature unlearned in ImageNet may be the discriminative features in them and some of their non-essential features are not removed during tuning in ImageNet. Thus, there is a slight CA loss and ASR increase after RVPT. Nevertheless, the in-dataset predictive features of real-world images seem quite similar, ensuring RVPT's cross-dataset defensive ability. Moreover, VPT's tendency to overfit the base dataset also contributes to the CA drop. Overall, *RVPT's performance on new datasets declines slightly but still outperforms CleanCLIP.*

**Performance on cross-domain transfer.** To evaluate the cross-domain defensive ability, RVPT was trained on ImageNet and tested on ImageNet-V2 [54], ImageNet-A [20], ImageNet-R [19], and ImageNet-S [61]. The result is shown in Table 6. *RVPT can generalize its defensive ability across different domains.* We also observe that RVPT will experience a loss of recognition and defense performance with domain shifts. In particular, it shows a significant CA drop on ImageNet-A, which reflects its effectiveness in focusing on in-dataset predictive features and ignoring other features, since ImageNet-A intentionally collects natural adversarial samples for ImageNet classifiers. Notably, unlike adversarial training [2], RVPT is not designed to defend against adversarial samples in ImageNet-A. The detailed comparison between RVPT and adversarial training is in Appendix F.

## 4.4 Ablation Studies

**Number of shots per class.** From Figure 3(a), we surprisingly find that only 1 shot can make the ASR of Blended and BadNet attacks nearly 0 and significantly decrease the ASR of BadCLIP and increasing the number of shots further enhances the defense. For BadCLIP, 8 shots are needed to bring the ASR below 5%. Nevertheless, with 32 shots, the defense effectiveness stops improving.

**Balancing factor $\alpha$.** We use $\alpha$ to balance the strengths of the CE loss and FR loss. From Figure 3(b), we observe that both ASR and CA decrease as $\alpha$ increases. This observation indicates that if the balancing factor is set too high, it will interfere with the encoding of in-dataset predictive features (the adversarial environment is too harsh for CE loss optimization), resulting in performance degradation. However, a relatively small $\alpha$ is sufficient to achieve very low ASR for every evaluated dataset.

**Visual context length.** We show RVPT's performance in Figure 3(c) with changing visual context length $b$. It indicates that both ASR and CA are optimal at a context length of approximately 25 to 50

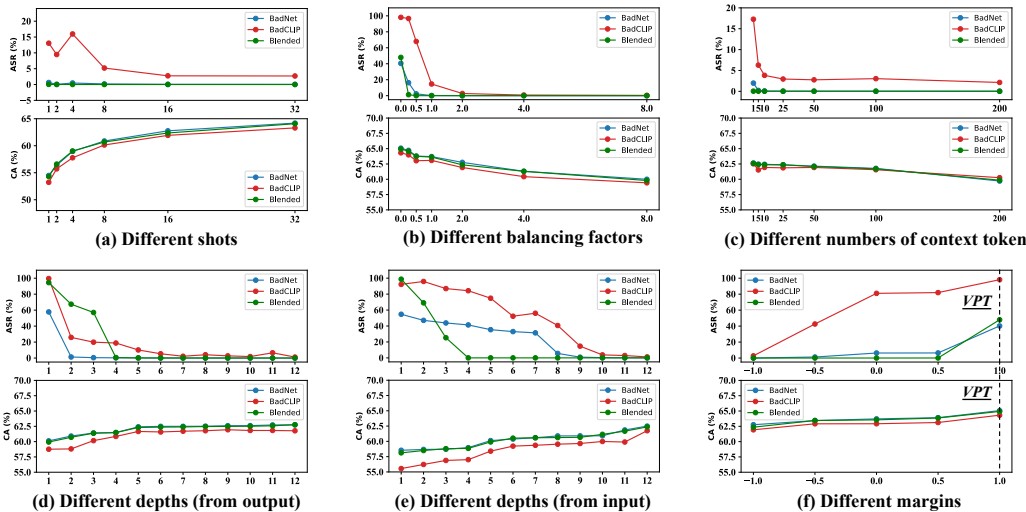

Figure 3: ASR and CA are evaluated when the **hyperparameters of shots**, $\alpha$, **context token length**, **depth**, and **margin** are changed in RVPT. In panel (f), the dashed lines represent the situation when the total FR loss is ablated. More ablation studies of visual encoder architecture and handcrafted text prompt can be referred to in Appendix E.

tokens. Beyond this optimal range, CA decreases due to overfitting caused by the increasing number of parameters. Moreover, when the length is 5, the ASRs of all attacks drop below 5%, indicating that tuning only 0.5‰ of parameters is sufficient to defend against all attacks, demonstrating the efficiency of our method. However, with increasing length, the defensive ability is bounded.

**Depth of the visual prompt.** Figure 3(d) shows the performance as the depth of the features into which the visual prompts are embedded is varied. It is important to note that "depth" here refers to the number of layers counted from the model's output feature. We also conduct experiments where prompts are stacked from the input feature, as shown in Figure 3(e). The comparison demonstrates that the later layers are more significant for backdoor robustness and recognition performance. We conjecture the reason is that the non-essential features encoded in the earlier layers tend to be sparse and intermixed with desirable features and repelling these features at an earlier stage is difficult and may inadvertently compromise the encoding of discriminative features, resulting in worse performance. Therefore, unless otherwise specified, we will count the depth from the output feature. Moreover, the results in Figure 3(d) indicate the improving performance as the number of the prompted layers increases. Particularly, when the depth reaches approximately six layers, both CA and ASR attain their optimal values.

**Margin of the repelling strength.** To evaluate the potential risks of over-repelling, we set a margin $m$ of FR Loss $\mathcal{L}_{\text{FRM}_i}$ for each sample $x_i$: $\mathcal{L}_{\text{FRM}_i} = \max(\mathcal{L}_{\text{FR}_i}, m)$. A greater $m$ means a lower level of repulsion, and $m = 1$ signifies no repulsion (which is *VPT*). The performance of RVPT of different $m$ is shown in Figure 3(f). We observe that both ASR and CA will increase with larger $m$. Therefore, for the purpose of achieving superior defense performance, we adopt full repulsion that optimizes the mean cosine similarity to -1.

## 4.5 Further Explorations

**Attention map and t-SNE.** We analyze the attention maps and t-SNE [58] plots of different models. In Figure 4, RVPT enables the encoder to focus on relevant bird features for clean images, while the backdoored model attends to irrelevant ones. For poisoned images, RVPT notices the trigger but avoids overemphasizing it, unlike the backdoored encoder. Figure 5 shows that backdoored CLIP and VPT models cluster around the trigger pattern, ignoring original image content [3]. RVPT, however, disrupts this cluster and restores poisoned image representations, reducing the influence of the trigger.

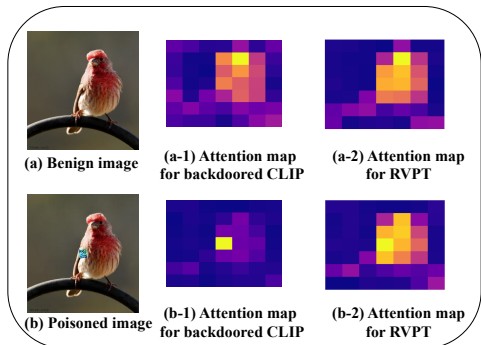

Figure 4: **Last-layer attention map** for (a) original (b) poisoned image in backdoored model (attacked by BadCLIP) and RVPT.

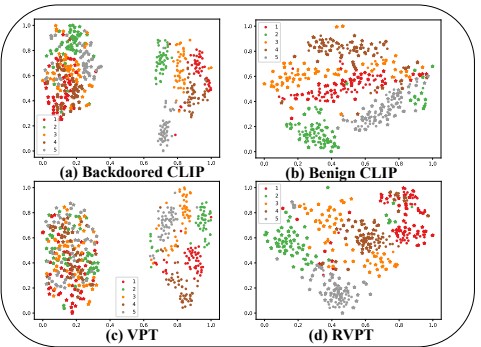

Figure 5: **The t-SNE [58] plots** for the representations of clean (dotted) and poisoned (star-shaped) images (attacked by BadCLIP).

**RVPT endows CLIP with backdoor robustness in various downstream tasks.** To further evaluate the impact of backdoored CLIP in real-world settings, we assess its performance on two downstream tasks: image captioning and text-to-image retrieval in Table 7. For image captioning, we use MSCOCO [36] and LLaVA-1.5 (13B) [37] equipped with backdoored CLIP ViT/L14. A successful attack is defined as generating a caption that includes the target class. For text-to-image retrieval, we use an enriched ImageNet-1K [10, 28] with backdoored CLIP (ViT/B32). We insert one poisoned image into the candidate set and if it appears in the top-10 matches for clean captions of the target class, we count it as a successful attack. Table 7 shows that backdooring CLIP during pre-training can transfer the backdoor behavior to the evaluated downstream tasks. Notably, our RVPT effectively mitigates backdoor effects while preserving clean performance, outperforming baselines [3, 26].

Table 7: **Results on image captioning (IC) and text-to-image retrieval (TR).** For IC, we evaluate backdoor robustness using ASR ($\downarrow$ %), defined as the percentage of generated captions that include the target class. Clean performance is measured using the BLEU score ($\uparrow$ %). For TR, ASR ($\downarrow$ %) is defined as the percentage of target-class captions where the poisoned image appears in the top-10 most similar images to evaluate the backdoor robustness. We also report the average rank ratio ($\downarrow$ %) of the poisoned image among target-class candidates. Clean accuracy ($\uparrow$ %) refers to the percentage of original captions correctly matched with their corresponding images (top-1 similarity).

| Attack | Image Captioning (IC) | | Text-to-Image Retrieval (TR) | | |
|---|---|---|---|---|---|
| | ASR ($\downarrow$ %) | BLEU | ASR ($\downarrow$ %) | *avg.* Rank Ratio ($\uparrow$ %) | CA (%) |
| *Clean model* | - | *7.65* | - | *95.83* | *69.34* |
| No defense | 95.12 | 6.22 | 74.67 | 9.20 | 73.24 |
| CleanCLIP | 23.03 | 4.32 | 13.33 | 25.54 | 68.95 |
| VPT | 81.21 | 4.64 | 22.67 | 26.80 | 71.88 |
| **RVPT** | **0** | 4.51 | **0** | **82.12** | 69.89 |

# 5   Conclusion

In this paper, we, for the first time, investigated how to utilize limited clean samples and tunable prompts to defend a backdoored multimodal contrastively pre-trained model (i.e., CLIP). We empirically showed that CLIP's low visual feature resistivity to input perturbations results in its vulnerability to backdoor attacks. To enhance this resistivity, we proposed a novel method called Repulsive Visual Prompt Tuning that guides CLIP to exclusively encode in-dataset predictive features. This method leverages our designed feature-repelling loss to minimize the mean cosine similarity between the prompted features and the original features across layers while optimizing cross-entropy loss to maintain clean accuracy on the downstream tasks. Comprehensive experimental results on multiple attacks and benchmark datasets demonstrated the effectiveness of our proposed method.

## Acknowledgment

This work is supported by the Big Data Computing Center of Southeast University. Haobo Wang is supported by the Fundamental Research Funds for the Central Universities (No. 226-2025-00004).

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

# Defending Multimodal Backdoored Models by Repulsive Visual Prompt Tuning

## Appendix

We summarize the Appendix as follows:

## A  Why Using VPT to defend backdoored CLIP?

### A.1  The drawback of fine-tuning the whole model

Fully fine-tuning can be done in two ways, both of which exhibit significant drawbacks in our setting.

**Fine-tuning whole CLIP with Info-NCE loss.** Since our primary downstream task is classification, we do not have downstream image-text pairs for optimization. We could use surrogate datasets such as CC3M, but as Table 8 shows, defense performance drops significantly. This degradation occurs because RVPT relies on identifying predictive features specific to the downstream dataset and pruning non-predictive ones. The mismatch between predictive features in CC3M and the actual downstream task leads to poor backdoor robustness.

**Fine-tuning whole CLIP with cross-entropy loss.** This requires adding an additional linear layer on top of the visual encoder for closed-set classification. However, this approach has several drawbacks: (i) It removes CLIP's open-set classification capability, severely limiting its generalization to other datasets. (ii) Given the limited labeled data, the large number of tunable parameters will cause overfitting, often resulting in significant catastrophic forgetting.

In summary, fine-tuning the whole CLIP with InfoNCE loss yields poor backdoor robustness and higher computational cost, while using cross-entropy loss slightly improves robustness but at the expense of much greater data requirements and loss of model generalization. Neither option provides a favorable trade-off between robustness, efficiency, and generalization.

Table 8: ASR (CA) performance of different fine-tuning methods with or without FR loss.

| Fine-tuning method | without FR Loss | with FR Loss | Open-set capability |
|---|---|---|---|
| ImageNet with CE Loss | 10.09 (60.71) | 0 (59.22) | NO |
| CC3M with Info-NCE Loss | 92.50 (54.50) | 18.23 (54.66) | YES |
| VPT on few-shot ImageNet (**preferred**) | 98.14 (64.32) | 2.76 (61.81) | YES |

## A.2 The reasons why we use VPT

As outlined above, full fine-tuning of CLIP presents significant drawbacks, either in terms of degraded robustness, excessive data requirements, or loss of open-set capabilities. To address these challenges, we adopt parameter-efficient fine-tuning methods (VPT) that:

- Leverage limited labeled downstream data effectively, aligning with RVPT's core mechanism of identifying task-relevant visual features.
- Preserve CLIP's open-set classification ability, which is critical for generalization across diverse tasks.

Specifically, since our goal is to enhance the perturbation resistivity of the CLIP's visual features, we inject learnable parameters into the visual modality, leading to our choice of VPT. This approach significantly improves downstream backdoor robustness using much less data while retaining CLIP's open-set capability, as demonstrated in Table 8.

## B More Results of Perturbation Resistivity of More Backdoored CLIP

In the pilot experiment, we evaluate the model's perturbation resistivity against 4 benign perturbations:

- Gaussian Noise with a standard deviation of 0.001.
- Gaussian Blur with a kernel size of 3 and variance of 5.
- Color Jitter with brightness of 0.5 and hue of 0.3
- Random Horizontal Flip.

The visualizations of these perturbations and trigger patterns can be found in Figure 6 and 9.

We evaluate the perturbation resistivity of four models:

- ViT/B32 of OpenAI CLIP.
- ViT/B32 of OpenAI CLIP tuned by VPT with 16 shots on ImageNet.
- ViT/B32 of OpenAI CLIP tuned by RVPT with 16 shots on ImageNet.
- ViT/B32 supervised exclusively on ImageNet.

The visual illustration for backdoored CLIP by BadNet and Blended are shown in Figure 7, 8 respectively. The results show that CLIP, even after VPT has generally much less PR value against various perturbations than traditionally supervised ViT, and RVPT can consistently increase its PR value across all scenarios.

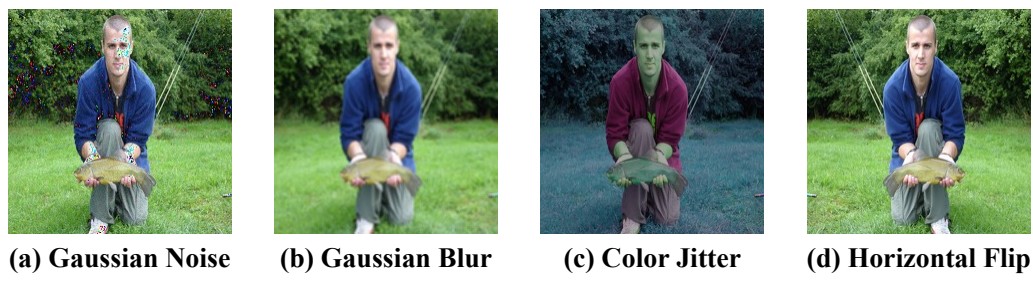

**(a) Gaussian Noise**     **(b) Gaussian Blur**     **(c) Color Jitter**     **(d) Horizontal Flip**

Figure 6: Visualization of the perturbations of various experimented attacks.

## C Detailed Settings

### C.1 Detailed Settings of Backdoor Attacks

In our experiments, we evaluate six prominent backdoor attack methods: BadNet [17], Blended [7], BadCLIP [35], ISSBA [31], WaNet [47], and TrojVQA [59]. Below, we provide a detailed description of each method:

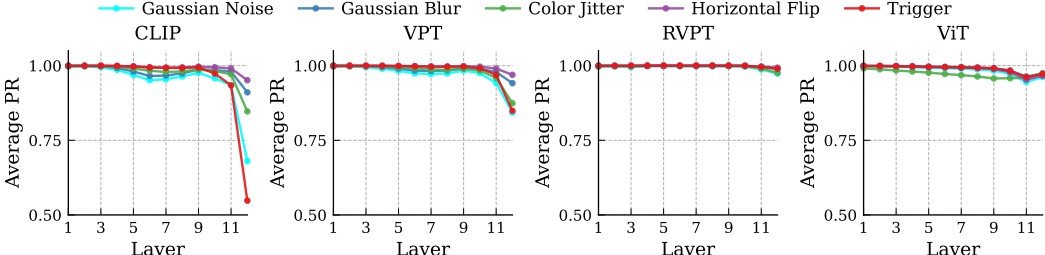

Figure 7: Average PR values of CLIP backdoored by BadNet.

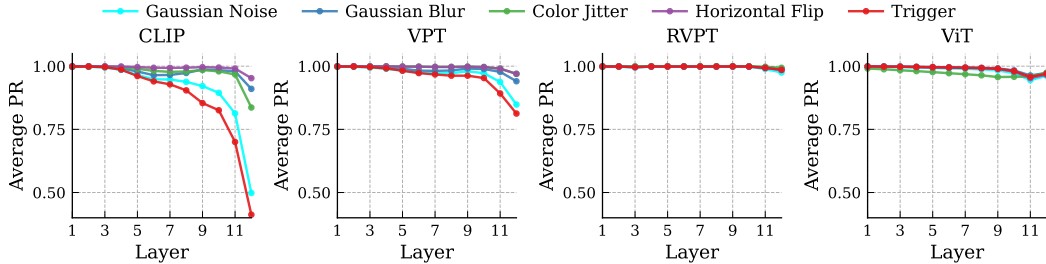

Figure 8: Average PR values of CLIP backdoored by Blended.

- BadNet, a foundational approach to backdoor attacks in deep learning, generates poisoned samples by embedding a small, randomly placed patch into images and modifying their labels to match the target class. In our implementation, the patch size is set to 16 pixels.

- Blended enhances the stealth of backdoor attacks by focusing on trigger design. This method linearly blends the trigger with the original image, creating an almost imperceptible backdoor. We adopt a blending ratio of 0.2 for the trigger to ensure minimal visibility to the human.

- BadCLIP targets CLIP models with a backdoor attack that optimizes visual trigger patterns using a dual-embedding guided framework. This method ensures that the attack remains undetectable by leveraging both text and image embeddings. We follow the parameter configurations in the original paper to implement BadCLIP.

- ISSBA introduces a highly covert attack by generating sample-specific triggers. These triggers are embedded into benign images using an encoder-decoder architecture, encoding a predefined string into the images. Moreover, the ciphertext is the string 'Stega!!'.

- WaNet proposes warping-based triggers to make the attack less noticeable to humans. Specifically, we use control grid size k = 224 and warping strength s = 1 and train models without the noise mode.

- TrojVQA proposes a dual-key backdoor attack that sets triggers on both modalities. To evaluate RVPT under backdoored generative VLMs, we use TrojVQA as the backdoor attack and use the following settings. Firstly, we optimize the visual triggers for the CLIP visual backbone with the description "This is a yellow banana.". Specifically, We set the patch size to 16×16 and located the patch in the middle of the image. Then, we randomly corrupt the text for poisoned samples into sentences that describe the target class of banana. Moreover, we add the trigger word "SUDO " in front of each sentence to make sure two keys for one backdoor. In the inference stage, the optimized trigger is added to the image, and "SUDO " is added to the beginning of the prompt.

We implement the attacks following [3]. Specifically, we employ the AdamW optimizer with an initial learning rate of 1e-6 for all the backdoor attack methods. The learning rate follows a cosine decay schedule over five epochs, and we set the batch size to 128. The visualization of each trigger pattern is in Figure 9.

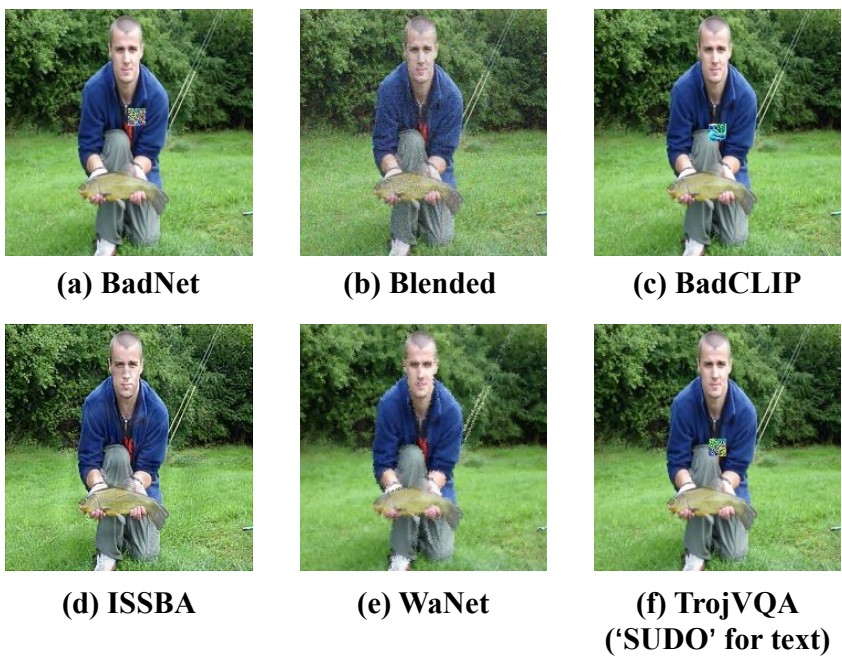

| (a) BadNet | (b) Blended | (c) BadCLIP |
| (d) ISSBA | (e) WaNet | (f) TrojVQA
('SUDO' for text) |

Figure 9: Visualization of the triggers of various experimented attacks.

## C.2 Detailed Settings of Backdoor Defenses

**CleanCLIP** CleanCLIP [3] provides a defense against backdoor attacks in multimodal contrastive learning by optimizing the integration of multimodal contrastive and unimodal self-supervised losses, relying on a limited amount of clean data. It is important to note that CleanCLIP utilizes ResNet-50 as the backbone of its visual encoder. In contrast, our work adopts the Vision Transformer (ViT-B/32) as the visual encoder, and we have accordingly adjusted the parameters to fit this architecture. Specifically, we randomly selected 250,000 image-text pairs from the CC3M dataset for fine-tuning. The learning rates were set to 5e-6 for BadNet, TrojVQA, Blended, and BadCLIP, while for WaNet and ISSBA on ImageNet-1K, they were set to 3e-6. We used a batch size of 64 and fine-tuned the models over 10 epochs. Notably, we did not focus solely on reducing the attack success rates by manipulating learning rates; instead, we ensured that the clean accuracy of the fine-tuned model was consistently maintained throughout.

**Linear Probe** Linear Probe [5] adds a tunable linear layer on top of the visual encoder, which seems efficient and effective in defending backdoor attacks. It does not truly purify the hidden trigger features. Consequently, it becomes ineffective against state-of-the-art multimodal attacks, whose trigger pattern aligns with the local image features [35]. This paper uses the same settings in [12] to adopt ViT-B/32 using the linear layer. Specifically, we use SGD as the optimizer and use the constant learning rate of 0.01 and momentum of 0.9. The batch size is 32, and the total training epoch is 40.

## C.3 Detailed Settings of Datasets

This paper assesses ASR and CA using three downstream datasets: ImageNet1K [10], Caltech101 [14], and OxfordPets [50]. For cross-domain evaluation, ImageNet-V2 [54], ImageNet-A [20], ImageNet-R [19], ImageNet-Sketch [61] are also evaluated. Additionally, CleanCLIP selects clean image-text pairs from CC3M [55] for fine-tuning the backdoored CLIP model. Below are detailed descriptions of these datasets:

- ImageNet1K includes 1,000 classes with over a million images, presenting a complex, large-scale dataset for image classification challenges.

- Caltech101 comprises 101 object classes and one background category, with each class containing between 40 and 800 images. It is widely used for evaluating models on fine-grained classification and image recognition.

- OxfordPets is a fine-grained dataset with 37 categories and approximately 200 images per class. The images display wide variations in scale, pose, and lighting.

- ImageNet-V2 was developed to assess model generalization under temporal shifts. It closely mirrors the original ImageNet's data collection and annotation processes, using new samples sourced from the Internet.

- ImageNet-A is designed to compile images that present notable challenges to deep learning models, often causing misclassification and thereby highlighting potential model vulnerabilities.

- ImageNet-R features "non-real" imagery, such as artworks, cartoons, and graphical representations, which differ significantly in visual style and form from the natural images found in the original ImageNet.

- ImageNet-Sketch contains hand-drawn sketches of the same categories as ImageNet, characterized by a focus on lines and shapes rather than colors and textures, presenting a distinct departure from photographic imagery.

- CC3M, or Conceptual Captions, includes around 3.3 million images paired with captions. Unlike other datasets with curated annotations, CC3M's image descriptions are sourced from web Alt-text attributes, thus offering a broader range of descriptive styles and contexts.

## D  Computational Expenses

We conducted the main experiments on eight single NVIDIA 3090 GPUs using a half-precision data type and recorded one training process in Table 9. The results indicate that our method substantially reduces computational costs, enhancing defense efficiency.

Table 9: Computational expense comparison between RVPT and CleanCLIP.

| Method | Training time | GPU memory | Parameters | Training samples |
|---|---|---|---|---|
| CleanCLIP | 3:53:02 | 17640 MB | 126 M | 250K |
| RVPT on ImageNet | 45:05 | 3382 MB | 0.34 M | 16 K |
| RVPT on OxfordPets | 2:19 | 1010 MB | 0.34 M | 0.6 K |

## E  More Experiments for Ablation Study

### E.1  Ablation Study of the Handcrafted Prompt

We experiment with the handcrafted prompt of "<CLS>" and "###### <CLS>" and show the result in Table 10. From the result, we can conclude that RVPT still successfully defends against backdoor attacks with different handcrafted prompts.

Table 10: Performance of RVPT with different handcrafted prompts. We report ASR ($\downarrow$ %), with CA ($\uparrow$ %) shown in parentheses. We can see the RVPT is stable regarding different handcrafted prompts.

| Handcrafted Prompt | ImageNet | | |
| | BadNet | blended | BadCLIP |
|---|---|---|---|
| "<CLS>" | 0.09 (62.28) | 0.03 (62.20) | 3.72 (61.73) |
| "###### <CLS>" | 0.05 (62.18) | 0.02 (61.96) | 3.78 (61.37) |
| "a photo of a <CLS>" | 0.05 (62.76) | 0.02 (62.36) | 2.76 (61.81) |

### E.2  Ablation Study of the Visual Encoder Architecture

To evaluate RVPT across different architectures, We poisoned different architectures of CLIP using the same hyper-parameters and show the result in Table 11. In particular, for the sake of fairness, we

keep the RVPT training setting for ViT-B/16 and ViT-L/14 the same as for ViT-B/32. The result shows that all attacks have been launched successfully, so we can conclude that RVPT still successfully defends against backdoor attacks in various visual encoder architectures.

Table 11: Performance of RVPT on attacked samples of ImageNet across different visual architectures. We report ASR ($\downarrow$ %), with CA ($\uparrow$ %) shown in parentheses. We can see the RVPT consistently successfully defends against the backdoor attacks in various visual encoder architectures.

| Method | BadNet | Blended | BadCLIP |
|---|---|---|---|
| *ViT-B/16* | | | |
| No defense | 99.61 (68.04) | 98.30 (67.84) | 99.67 (68.09) |
| Linear Probe | 7.11 (67.52) | 0.21 (67.30) | 97.73 (67.60) |
| RVPT | **0.66** (68.51) | **0.02** (68.32) | **3.38** (68.42) |
| *ViT-L/14* | | | |
| No defense | 98.76 (73.88) | 99.02 (74.25) | 99.87 (74.68) |
| Linear Probe | 23.48 (72.31) | 0.65 (72.35) | 99.35 (72.35) |
| RVPT | **2.08** (73.60) | **0.05** (73.26) | **0.59** (74.04) |

# F    Discussion about the Difference between RVPT and Adversarial Training

As illustrated in Section 3, RVPT adopts the FR loss to adversarially repel features to impede the learning process, which partly shares some similarity with adversarial training (AT) [2]. While this approach shares some superficial similarities with adversarial training (AT) [2], there are fundamental differences between the two methodologies.

First, AT generates the predictive yet brittle features [23] via gradient-based attacks [16, 44], and then unlearns them with correct labels. (first pick bad features then unlearn) In contrast, RVPT first actively scrambles the feature representations and then learns the most predictive features. (first scramble features then pick good ones)

As a result, AT prevents the model from encoding the predictive, brittle features, while RVPT encourages the model to encode the in-dataset predictive features. In the training process of RVPT, there are no adversarial features unlearned. Therefore, it cannot defend against adversarial attacks. Table 12 compares the performance of RVPT and a representative adversarial training method of CLIP under adversarial attack. However, there are non-predictive features or out-of-dataset features unlearned, which ensures RVPT to defend against backdoor attacks.

Table 12:  Accuracy of RVPT and adversarial Prompt Tuning [71] of adversarial samples on ImageNet. The attack is PGD-10, which maximizes the cross-entropy loss with a budge of 8/255.

| | No defense | RVPT | Adversarial Prompt Tuning |
|---|---|---|---|
| Accuracy | 4.37 | 4.10 | 13.97 |

# G    Broader Impacts

**Positive Impact**

- **Improved security:** The proposed Repulsive Visual Prompt Tuning (RVPT) significantly improves the robustness of multimodal models (e.g., CLIP) against backdoor attacks, thereby enhancing the security of AI systems deployed in various real-world applications. This defense mechanism helps protect systems from adversarial manipulation, improving their reliability in tasks such as image captioning and text-to-image retrieval.

- **Cost-effective defense:** RVPT only requires a small number of clean samples and tunes only a few parameters, making it an efficient and low-resource method compared to traditional full-model fine-tuning approaches. This could lower the barrier for implementing robust defenses in practical AI systems.

- **Cross-domain generalization:** RVPT shows promise across different datasets and domains, which enhances its applicability to a broad range of tasks and platforms, making it a versatile defense tool in multimodal AI systems.

**Negative Impact**

- **Potential misuse:** While RVPT is designed to defend against backdoor attacks, its approach could potentially be misused to manipulate AI systems in unintended ways. For example, adversaries could leverage similar techniques to create or enhance stealthy backdoor attacks, making the technology a double-edged sword.

- **Overfitting and trade-offs:** The need for some fine-tuning and the requirement for a clean dataset means that the model might not be fully adaptable to all scenarios, leading to potential overfitting to specific tasks or datasets. This could limit its robustness in highly dynamic or unseen environments.

- **Computational costs:** While RVPT is more efficient than full-model fine-tuning, it still requires additional computational resources for training and testing, which could be a challenge for low-resource environments or in large-scale deployments.

# H   Limitations

- **Limited dataset generalization:** Although RVPT performs well across multiple datasets, there are instances where performance drops slightly, particularly when the training data and test data are from different domains (e.g., ImageNet vs. Caltech101 or OxfordPets).

- **Dependence on clean data:** RVPT's reliance on a small number of clean samples limits its applicability in scenarios where clean data is scarce or unavailable. This could restrict the use of RVPT in certain contexts where adversarially poisoned data is prevalent.

- **Vulnerability to new attack strategies:** While RVPT is effective against known attacks, it might not fully defend against novel attack strategies that do not rely on traditional triggers. As with all defense mechanisms, its robustness against future threats is not guaranteed.

