# OpenReview forum: "Defending Multimodal Backdoored Models by Repulsive Visual Prompt Tuning"
_NeurIPS.cc/2025/Conference — NeurIPS 2025 poster_

### Official Review · Reviewer_aLX9 · 2025-06-09

**Clarity:** 4
**Significance:** 3
**Originality:** 3
**Rating:** 4
**Confidence:** 3

**Summary:**

CLIP aligns visual and textual representations in the pre-training stage and serves as a powerful base for multimodal tasks. CLIP has been shown vulnerable to backdoored attacks, which inject a small proportion of poisoning data to make the model behave normally without the trigger, but reliably predict the target class with the trigger. The paper tries to “clean” a backdoored CLIP by training with a downstream clean dataset. The proposed RVPT optimizes some newly added layer-wise features to minimize the cosine similarity between the forwarded features and the original clean CLIP features. This proposed feature-repelling loss works with the cross-entropy loss to increase robustness against input perturbations while preserving utility.

**Questions:**

See weaknesses

**Ethical Concerns:**

["NO or VERY MINOR ethics concerns only"]

**Final Justification:**

Thanks for the rebuttal, which addresses most of my concerns. Thus, I maintain my positive scores. I encourage the authors to include the insightful discussions on "Why is VPT used" in the paper.

**Limitations:**

Yes

**Quality:**

4

**Strengths And Weaknesses:**

Strengths
1. The proposed Perturbation Resistivity is an intuitive and straightforward metric to constrain the image features to be robust against perturbations. For embeddings, the direction tends to indicate rich information compared to the magnitude.
2. The threat model is clearly stated. The attacker poisons the clean CLIP, and the defender uses a small clean set to reverse the backdoor effect by additional training. It requires less clean data as it starts from a well-functioning backdoored model, and the only task here is to remove the backdoor.
3. The experiments are comprehensive and show strong effectiveness of RVPT, which achieves near 0% attack success rates using various attack methods, significantly surpassing existing baselines. The authors perform extensive study on the generalization and hyperparameters of the defense.

Weaknesses
1. It is unclear to me why VPT is used. The optimization objective makes sense, but why do the authors choose to optimize additional features as VPT. The introduction mentions that fine-tuning the whole model is “resource-intensive or even unrealistic”, but I am curious if that will lead to better performance, and what is the trade-off.
2. I don’t understand the ablation study on emerging target class, mostly because of the term “tuning dataset”. If it stands for the attacker’s poisoning dataset, how can an attacker poison the model without showing the in-class samples? If it stands for the defender’s downstream dataset, only one term should be used to describe it with proper explanations.
3. The discussion about the connection to adversarial attacks is lacking. How can the proposed loss inspire defenders against test-time attacks for images where reducing model’s sensitivity to input perturbations is also the key?

---

> ### Author Rebuttal · Authors · 2025-07-31
>
> Thank you for your valuable comments!
>
> ## W1: Why is VPT used?
> Thanks for this insightful question! Before addressing why we use VPT, we would like to clarify the trade-offs involved when fine-tuning the entire CLIP model and explain why full fine-tuning is not preferred in our context.
>
> ### The trade-off when fine-tuning the whole model
>
> Full fine-tuning of CLIP can be performed in two main ways:
>
> - Using the InfoNCE loss on an image-text dataset.
> - Using the cross-entropy (CE) loss on an image-label classification dataset.
>
> **Fine-tuning whole CLIP with Info-NCE loss.** Since our primary downstream task is classification, we do not have downstream image-text pairs for optimization. We could use surrogate datasets such as CC3M, but as Table 1 shows, defense performance drops significantly. This degradation occurs because RVPT relies on identifying predictive features specific to the downstream dataset and pruning non-predictive ones. **The mismatch between predictive features in CC3M and the actual downstream task leads to poor backdoor robustness.**
>
> **Fine-tuning whole CLIP with cross-entropy loss.** This requires adding an additional linear layer on top of the visual encoder for closed-set classification. However, this approach has several drawbacks:
> - It **removes CLIP's open-set classification capability**, severely limiting generalization to other datasets.
> - Given the limited labeled data, the large number of tunable parameters causes overfitting, resulting **catastrophic forgetting**.
> - Although it can eliminate backdoors (see Table 1), it demands **much more data** to avoid overfitting and sacrifices the model's generalization ability, making it an unfavorable option.
>
>
> **Table 1: ASR (CA) Performance of different fine-tuning methods with or without FR loss.**
> |Fine-tuning method|without FR Loss| with FR Loss|Open-set classification capability|
> |--|--|--|--|
> |Fine-tuning on full ImageNet using CE Loss| 10.09 (60.71)|0 (59.22)|NO|
> |Fine-tuning on CC3M using info-NCE Loss|92.50 (54.50)|18.23 (54.66)|YES|
> |Visual prompt tuning on few-shot ImageNet **(preferred)**|98.14 (64.32)|2.76 (61.81)|YES|
>
>
> To summarize, if we fine-tune the whole CLIP with Info-NCE loss, it will all become worse with heightened computational cost and less backdoor robustness (no trade-offs). If we use CE loss, we can only gain a little backdoor robustness at the sacrifice of much more data demand and deprivation of model generalization ability.
>
> ### The reasons why we use VPT
>
> As outlined above, full fine-tuning of CLIP presents significant drawbacks, either in terms of degraded robustness, excessive data requirements, or loss of open-set capabilities. To address these challenges, we adopt parameter-efficient fine-tuning methods that:
> - Leverage limited labeled downstream data effectively, aligning with RVPT's core mechanism of identifying task-relevant visual features.
> - Preserve CLIP's open-set classification ability, which is critical for generalization across diverse tasks.
>
> Specifically, since our goal is to enhance the perturbation resistivity of the CLIP's **visual features**, we inject learnable parameters into the visual modality, leading to our choice of VPT. This approach significantly improves downstream backdoor robustness using much less data while retaining CLIP's open-set capability, as demonstrated in Table 1.
>
>
> ## W2: The explanation of the ablation study on the emerging target class
> Thanks for pointing out this ambiguity. We apologize for the confusion caused by the term "tuning dataset" and will clarify our terminology in the revised manuscript.
> The reviewer is correct: the "tuning dataset" refers to the **defender's downstream dataset**.
> We will revise Section 4.3 to explicitly use the term "**defender's clean tuning set**" and clarify this experimental process to prevent any future misinterpretation.
>
> ## W3: RVPT's connection to the defense against adversarial attacks
> Thanks for this excellent and thought-provoking question regarding the broader implications of our work. We discuss this connection in more detail in Appendix E, but we are happy to elaborate further here.
>
> RVPT is specifically designed to reduce a model's sensitivity to **non-predictive** input perturbations, which are characteristic of backdoor triggers. In contrast, adversarial perturbations are typically **predictive and brittle**, as they are carefully optimized to manipulate model outputs while remaining imperceptible [1].
>
> Because RVPT is explicitly designed to preserve predictive features and repel non-predictive ones, its defense mechanism is **not directly applicable to adversarial attacks**, which often exploit predictive but fragile features. Therefore, RVPT is not guaranteed to defend against adversarial perturbations.
>
>
> However, as your suggestion insightfully points out, there is a conceptual overlap that can be built upon. Specifically, the challenge in extending RVPT to adversarial defense lies in disentangling brittle adversarial predictive features from robust predictive features. Simply eliminating all predictive components would lead to unacceptable degradation on benign inputs. To address this, we envision a future extension of RVPT that incorporates an anchor-based mechanism: For a given adversarial example, we retrieve or synthesize a semantically similar clean "anchor" image, which shares robust, task-relevant features. We then define a repulsive loss between the feature representations of the adversarial and anchor images. This loss is designed to repel the components that are unique to the adversarial image, while preserving shared features that reflect the true semantic content. Such a mechanism would allow us to selectively suppress brittle predictive features introduced by the adversarial noise, without harming generalization or clean accuracy.
>
> We see this as an exciting and promising direction for future research in defending against test-time adversarial attacks using feature-level control strategies.
>
>
> ## References
> [1] Adversarial examples are not bugs, they are features, NeurIPS 2019.

---

> > ### Comment · Reviewer_aLX9 · 2025-08-02
> > **Thanks for the rebuttal**
> >
> > Thanks for the rebuttal, which addresses most of my concerns. Thus, I maintain my positive scores. I encourage the authors to include the insightful discussions on "Why is VPT used" in the paper.

---

> > > ### Author Response · Authors · 2025-08-02
> > >
> > > We sincerely thank you for maintaining your positive evaluation and for your encouraging feedback. We will incorporate the suggested discussion on “Why VPT is used” in the final version of the paper, as we agree it is important and adds valuable context.

---

### Official Review · Reviewer_w7GP · 2025-06-22

**Clarity:** 3
**Significance:** 3
**Originality:** 2
**Rating:** 4
**Confidence:** 3

**Summary:**

This paper addresses the significant vulnerability of multimodal contrastive learning models like CLIP to backdoor attacks. It propose that CLIP’s susceptibility arises from  its tendency to encode features beyond in-dataset predictive patterns. To mitigate this, they introduce a new defense mechanism called Repulsive Visual Prompt Tuning (RVPT). The results show that RVPT is highly effective, drastically reducing the ASR while preserving clean accuracy and being significantly more efficient than existing methods.

**Questions:**

1. Would RVPT still maintain strong defense capabilities if the number of clean downstream samples were significantly reduced, or in extreme cases, absent?

2. Figure 3(b) indicates a trade-off controlled by the balancing factor $\alpha$, showing both CA and ASR decrease as $\alpha$ increases. In real-world deployment scenarios, a defender might only have access to a new small clean dataset and a backdoored model, without means to accurately measure the ASR. Under these practical constraints, how would you recommend defenders set the value of $\alpha$?

3. What are the computational costs introduced by the feature-repelling mechanism in comparison to APT baseline?

4. Could we inject learnable prompts on both the text and image branch, similar to prior work [1, 2].

[1] MaPLe: Multi-modal Prompt Learning, CVPR 2023.

[2] TAPT: Test-Time Adversarial Prompt Tuning for Robust Inference in Vision-Language Models, CVPR 2025.

**Ethical Concerns:**

["NO or VERY MINOR ethics concerns only"]

**Final Justification:**

Thank you to the authors for their detailed rebuttal. After reading rebuttal, I tend to keep my rating.

**Limitations:**

Detailed computational overheads (time, resources, latency) are missing, which could hinder practical adoption in resource-constrained scenarios.

**Quality:**

3

**Strengths And Weaknesses:**

Paper Strengths:

1. The authors evaluate their approach on multiple benchmarks and conduct extensive ablation studies.

2. The experimental setup is described in a detailed and thorough manner.

Weaknesses:

1. The evaluation does not consider potential adaptive attacks explicitly tailored to RVPT’s defense strategy.

2. The method introduces several new hyperparameters, most notably the balancing factor, the prompt depth, and the prompt length. The ablation studies show that the model's performance is sensitive to these choices.

---

> ### Author Rebuttal · Authors · 2025-07-31
>
> Thank you for your valuable comments!
>
> ## W1: Potential adaptive attacks
> Thanks for the question. We respectfully hold that crafting an **effective adaptive attack against RVPT is highly challenging under most realistic threat models**.
>
> The core reason lies in the **temporal and informational gap** between the poisoning stage and the application of RVPT. Specifically, CLIP is assumed to be poisoned during pre-training, whereas RVPT is applied post hoc, using private downstream data for adaptation. For an adaptive attack to succeed, the adversary would need foreknowledge of the user's downstream data during the upstream pre-training phase, which is a highly unrealistic assumption. Even if the downstream data becomes accessible and an adaptive attack is crafted successfully, its effectiveness would likely not transfer to other downstream tasks, where CLIP is fine-tuned with different data under RVPT, severely limiting the threat scope.
>
> Furthermore, another reason for being impractical to design an adaptive attack is the **mechanism of RVPT**: it filters non-predictive features and only retains predictive features in a clean downstream dataset.
> This makes it non-trivial to design a trigger that can be encoded by backdoored CLIP as predictive features in the clean downstream dataset. In fact, we experimented with using a universal adversarial perturbation [1] as a trigger to poison CLIP, but this strategy failed after applying RVPT, reinforcing the difficulty of designing such attacks.
>
> We believe that constructing highly specialized adaptive attacks against RVPT could be an interesting direction for future research, but under current assumptions, such attacks appear highly challenging and limited in scope.
>
>
> ## W2: Performance sensitive to hyperparameters
>
> Thanks for your concern regarding hyperparameter sensitivity. However, we believe that while RVPT does introduce several hyperparameters, we find that they are **straightforward to configure** due to their robustness across a wide range of values.
>
> - For prompt depth, our ablation study shows that both clean accuracy and defense effectiveness improve and then saturate with increasing depth. This means there is **no trade-off**, and practitioners can simply select the maximum feasible depth without risk of performance degradation.
> - For the balancing factor ($\alpha$) and prompt length, sensitivity primarily arises at very low values, where the defense has not yet fully activated. As our results demonstrate, once these parameters exceed modest thresholds (e.g., $\alpha \geq 2.0$, prompt length $\geq 25$), RVPT **consistently achieves near-zero ASR while maintaining high CA**.
>
> In this context, the observed "sensitivity" is not a practical limitation but rather a clear and interpretable indicator of an effective operating range. Defenders can therefore select hyperparameters confidently from this broad, stable region to achieve strong and reliable security.
>
>
> ## Q1: Performance with reduced number of samples
> Thanks for raising this important question regarding the data requirements of our method.
>
> **Performance with few samples.** Our experiments, detailed in Figure 3(a), demonstrate that RVPT remains highly effective even in extremely low-data regimes. For traditional vision-only backdoor attacks such as BadNet and Blended, just one clean sample per class (1-shot) suffices to achieve robust defense. For the more challenging state-of-the-art multimodal attack: BadCLIP, RVPT requires only eight samples per class (8-shot) to reduce the ASR below 5%. This represents significant data efficiency compared to previous post-training defenses: for example, RVPT defends CLIP using only 6% CleanCLIP's data on ImageNet and 0.2% CleanCLIP's data on OxfordPets.
>
> **Performance in zero-Shot case.** We believe that RVPT is not directly applicable when no clean data is available by design, since The core principle of our method is to use a small set of trusted downstream samples to identify task-relevant visual features. By identifying predictive features, RVPT can effectively detect and suppress extraneous features, including potential backdoor triggers. Therefore, without any such reference samples, RVPT cannot distinguish predictive from non-predictive features, and thus cannot perform the defense. Extending multimodal backdoor post-training defenses to a fully zero-shot setting remains an important and valuable direction for future research.
>
> ## Q2: The choice of $\alpha$
> Thanks for the insightful feedback! We have observed that the ASR is highly sensitive to lower values of the balancing factor $\alpha$, but rapidly approaches zero as $\alpha$ increases. In contrast, the CA is much less sensitive and decreases gradually and gracefully. This behavior allows defenders to adopt a conservative strategy by selecting a larger $\alpha$ to maximize robustness, with minimal impact on clean performance. Alternatively, defenders can tune $\alpha$ based on an acceptable CA threshold, which is measurable during deployment. Specifically, we recommend the following two strategies for setting $\alpha$:
>
>
> **Strategy 1: Use a recommended conservative default $\alpha$.** Based on empirical results across various attack types, we suggest a default value of $\alpha = 2.0$. As shown in our main experiments, this value consistently reduces ASR to near zero for all evaluated attacks while only incurring a minor CA decrease of approximately 1–2% relative to $\alpha=0$.
>
> **Strategy 2: Tune using a clean validation set.** If a defender wants to tune $\alpha$ for their specific task, they can follow this simple procedure:
> 1. Start with a small $\alpha$ (e.g., 0.5).
> 2. Incrementally increase $\alpha$ (e.g., to 1.0, 1.5, 2.0, ...).
> 3. At each step, measure the CA on the clean validation set.
> 4. Select the largest value of α for which the drop in CA is still acceptable.
>
> This approach enables defenders to push defense strength as high as possible within their performance budget. Since ASR decreases more rapidly than CA, the chosen $\alpha$ is very likely to have neutralized any backdoor effectively.
>
> In summary, while defenders cannot directly measure ASR, they can reliably measure CA. The predictable and gentle decline in CA allows them to choose a value for $\alpha$ that strongly prioritizes security at a minimal and quantifiable cost to performance.
>
> ## Q3: The added computational cost of FR loss
>
> Thanks for your questions! We would like to clarify that the feature-repelling mechanism does not introduce a new CLIP model. The output of the original backdoored CLIP can still be obtained by using features without the visual prompts.
>
> In Table 1, we compare the computational costs of RVPT, its baseline VPT (which is RVPT without the feature-repelling mechanism), and a state-of-the-art post-training defense, CleanCLIP.
>
> **Table 1: Computation cost of RVPT, VPT (RVPT without feature-repelling mechanism), and CleanCLIP.**
> | Method | Training time (in hour) | Used GPU memory | Tunable Parameters |
> | -------- | -------- | -------- |-------- |
> |**VPT**|   0.72   |   3382 MB   |  0.34 M   |
> |**RVPT**|  0.95 (+31.9%)    | 3382 MB  (+0%)   |  0.34 M  (+0%) |
> |**CleanCLIP**|  3.88  (+454.2%)  |  17640 MB (+421.6%) |   126 M (+36958.8%) |
>
> As shown in Table 1, RVPT introduces only a 31.9% increase in training time compared to the standard VPT baseline. Since VPT itself is a parameter-efficient fine-tuning method, this marginal overhead is negligible—especially when compared to the substantially higher resource demands of CleanCLIP. This minimal computational cost highlights the practicality and efficiency of RVPT for real-world deployment.
>
> ## Q4: Injection of learnable prompts into the multimodal branch
> Thanks for the insightful suggestion. Following your advice, we conducted an experiment incorporating the MaPLe [2] architecture, which uses learnable prompts in both the text and image branches. The results are summarized in Table 2. In this setup, the visual prompts were optimized using both cross-entropy loss and our feature-repelling loss, while the textual prompts were optimized using only the cross-entropy loss.
>
> **Table 2: Ablation study on the ASR (CA) performance of the feature-repelling mechanism in the architecture of MaPLe.**
> | Method | BadNet | Blended | BadCLIP |
> | -------- | -------- | -------- |-------- |
> |MaPLe|   61.72 (63.77)   |  60.19 (63.92)    |  98.85 (62.67)   |
> |MaPLe + feature-repelling mechanism|   **1.02** (59.87)   | **0.01** (59.42)     |  **1.24** (58.60)   |
>
> As shown in Table 2, optimizing prompts on both branches without the feature-repelling mechanism does not significantly improve backdoor robustness. However, when the visual prompt is further optimized with the feature-repelling loss, the robustness improves substantially. This finding reinforces our core claim: the effectiveness of RVPT primarily arises from operations within the visual encoder. Our feature-repelling mechanism effectively neutralizes trigger patterns in the visual feature space, making the defense largely independent of the textual branch.
>
> ## References
> [1] Universal adversarial perturbations, CVPR 2017.
> [2] MaPLe: Multimodal Prompt Learning, CVPR 2023.

---

### Official Review · Reviewer_3JXv · 2025-06-24

**Clarity:** 3
**Significance:** 3
**Originality:** 3
**Rating:** 4
**Confidence:** 3

**Summary:**

This paper introduces a novel backdoor attack defense method, Repulsive Visual Prompt Tuning (RVPT), which adversarially pushes the encoded features from deeper layers away from those of the backdoored model while still optimizing performance on downstream tasks. The RVPT method enhances CLIP’s visual feature robustness against various backdoor attacks and under diverse conditions.

**Questions:**

1. I wonder whether the proposed feature repelling loss can be applied to models that are not contrastively trained like CLIP.
2. It is unclear how the method behaves when applied to an unpoisoned model. Does RVPT improve robustness even in the absence of a backdoor, or does it unintentionally degrade the model's performance on clean inputs?
3. I cautiously hypothesize that CLIP’s strong zero-shot performance may partly stem from its ability to capture a wide range of off-dataset or non-predictive features. I am curious whether the RVPT-processed CLIP model maintains this generalization capability, particularly in zero-shot settings.

**Ethical Concerns:**

["NO or VERY MINOR ethics concerns only"]

**Final Justification:**

This paper proposes RVPT, a novel backdoor defense method that repels encoded features from those of a backdoored model while preserving downstream performance, thereby enhancing CLIP’s robustness across diverse backdoor attacks and settings. While the BLEU score drop for LLaVA-1.5 is notable, the authors’ explanations addressing my other concerns are convincing, and I therefore maintain a positive stance.

**Limitations:**

yes. They are in Appendix.

**Paper Formatting Concerns:**

No Formatting Concerns

**Quality:**

3

**Strengths And Weaknesses:**

Strengths:
1. The paper conducts extensive experiments under diverse conditions, including various datasets, various types of backdoor attacks, and cross-domain settings.
2. The authors provide attention map and t-SNE analyses, offering visual evidence that the proposed RVPT method effectively removes backdoored features.
3. The paper is clearly written and easy to follow.

Weaknesses:
1. This paper shows extensive experiments on various conditions, but they are all conducted on the CLIP architecture. As the title refers to multimodal backdoored models, I recommend to to demonstrate robustness on other multimodal models such as BLIP.
2. Limited applicability due to assumption of a Poisoned CLIP. However, in most real-world scenarios, such prior knowledge is unavailable; the poisoning status of the model is typically unknown. This assumption limits the practical applicability of the approach.

---

> ### Author Rebuttal · Authors · 2025-07-31
>
> Thank you for your insightful reviews!
>
> ## W1 & Q1: More multimodal architectures
> Thanks for this excellent suggestion. We believe that the feature repelling (FR) loss can be applied to multiple multimodal models, but it will only work on contrastive pre-trained models, since FR loss enhances the model's backdoor robustness by eliminating irrelevant features that do not contribute to cross-entropy (CE) loss. To compute CE loss, we have to adopt contrastive pre-trained models since their outputs are aligned into one shared space, which allows CE loss to be computed directly via cosine similarity during fine‑tuning. Therefore, FR loss probably remains ineffective for models trained without contrastive pre‑training, since their representations cannot be meaningfully compared through dot products.
>
> Importantly, most prevailing multimodal models, including recent large vision‑language models (LVLMs), adopt visual encoders that are pre-trained contrastively. This design of LVLMs has become the dominant trend for future multimodal systems, as contrastive pre‑training unifies the visual and textual embedding space, making it easier to align LLM's embedding space with visual encoders.
> Therefore, providing backdoor robustness to contrastively pre‑trained multimodal models is not only practical but also highly impactful for the field. For example, we defend the contrastively pre‑trained visual encoder of LLaVA‑1.5 using RVPT. As shown below, RVPT substantially reduces the attack success rate (ASR) in the image captioning task, while preserving a competitive BLEU score (stands for model utility).
>
> **Table 1: ASR (defense effectiveness) and BLEU (model utility) performance of multiple methods when defending LLaVA 1.5 in the image captioning task.**
>
> | Method       | ASR (↓ %) | BLEU (↑) |
> |--------------|-----------|----------|
> | Clean model  | -         | 7.65     |
> | No defense   | 95.12     | 6.22     |
> | CleanCLIP    | 23.03     | 4.32     |
> | VPT          | 81.21     | 4.64     |
> | **RVPT**     | **0.00**  | 4.51     |
>
> This demonstrates that RVPT can effectively mitigate backdoor risks in real downstream applications while maintaining utility, reinforcing its importance for the broader LVLM ecosystem.
>
>
> ## W2: Limited applicability
>
> Thanks for raising this critical point. We agree that, in most real-world scenarios, the poisoning status of a pre-trained model is unknown. However, in safety-critical downstream applications, practitioners often require absolute assurance of model integrity, meaning that the possibility of an unknown backdoor is unacceptable, and the **worst-case scenario** (i.e., the model is backdoored) must be assumed. For instance, a military drone relying on a pre-trained model for target recognition could suffer catastrophic consequences if a hidden backdoor causes it to misidentify an ally as an enemy. In such settings, even a small risk of manipulation is intolerable. In fact, there is already a paper demonstrating that **OpenAI CLIP has been backdoored** [1], underscoring the urgency and real-world relevance of our research.
>
> Importantly, RVPT provides strong backdoor robustness with minimal impact on clean data performance, making it a practical and necessary solution for deploying third-party foundation models in trust-critical environments.
>
>
> ## Q2: Performance of RVPT with unpoisoned model
> Thanks for the concern. RVPT is primarily designed to defend against backdoor attacks by identifying and suppressing malicious trigger features. When applied to an unpoisoned model, RVPT is not expected to offer direct benefits for backdoor robustness, simply because there is no backdoor behavior to mitigate.
>
> However, it is important to note that **RVPT does not degrade model performance on clean inputs**. In fact, it can slightly improve it. As shown in Table 2, RVPT improves or maintains clean accuracy compared to both Zero-shot CLIP. This suggests that RVPT can act as a benign regularizer, even in the absence of backdoor triggers:
>
> **Table 2: Clean accuracy when the base CLIP is unpoisoned. Both RVPT and CleanCLIP fine-tune the unpoisoned CLIP.**
> | Method | ImageNet | Caltech101 |OxfordPets |
> | -------- | -------- | -------- |--|
> | Zero-shot CLIP     | 62.01   | 91.46     |86.42 |
> | CleanCLIP     | 60.58     | 91.35     |83.79|
> | RVPT     | **62.95**    | **94.51**     |**89.67**|
>
>
> ## Q3: Generalization capability of RVPT
> Thanks for raising this important point. You are absolutely correct that CLIP’s strong zero-shot performance likely arises from its ability to leverage a broad spectrum of features, including some that may be non-predictive or off-distribution. As shown in Table 3, this broad feature reliance explains the slight degradation in generalization ability observed after applying RVPT.
>
> However, we argue that this modest trade-off in generalization is acceptable given the significant gains in backdoor robustness. Moreover, this phenomenon is not unique to RVPT: many prompt learning and fine-tuning methods for CLIP also exhibit reduced zero-shot transferability due to overfitting to limited supervision signals. RVPT’s behavior is consistent with this trend. By explicitly suppressing predictive-but-brittle features, RVPT may slightly reduce generalization, but it does so to prioritize safety and robustness.
>
> **Table 3: Average ASR and zero-shot clean accuracy drop of defense methods. CLIP is trained on ImageNet and tested on Caltech101 and OxfordPets for the cross-dataset scenario and ImageNet-R, S, V2 for the cross-domain scenario.**
>
> | Method |  cross-dataset ASR ($\downarrow$CA drop) |cross-domain ASR ($\downarrow$CA drop) |
> | -------- | -------- | -------- |
> | CleanCLIP    | 14.98 ($\downarrow$ 1.79)     | 40.09 ($\downarrow$ 5.15)     |
> | RVPT    | **1.77** ($\downarrow$ 1.76)     | **1.42** ($\downarrow$ 5.05)     |
>
>
> To sum up, while RVPT does slightly trade off generalization for robustness (as is common in prompt learning), the degradation remains modest and acceptable. Importantly, RVPT still preserves competitive zero-shot performance across datasets and domains.
>
> ## References
> [1] Detecting backdoor samples in contrastive language image pretraining, ICLR 2025.

---

### Official Review · Reviewer_g6BJ · 2025-07-02

**Clarity:** 3
**Significance:** 2
**Originality:** 3
**Rating:** 3
**Confidence:** 4

**Summary:**

The paper introduces Repulsive Visual Prompt Tuning (RVPT), a lightweight defense mechanism against multimodal backdoor attacks targeting CLIP. The central idea is to improve perturbation resistivity, a metric proposed by the authors to measure the sensitivity of CLIP's embeddings to input noise. This is achieved by combining a cross-entropy loss with a novel feature-repelling loss, which discourages the encoding of irrelevant or potentially malicious features. The method updates only a small subset of the model's parameters and uses a clean dataset to mitigate backdoor vulnerabilities without requiring full model fine-tuning. Experiments across various attacks, datasets, and settings show substantial reductions in attack success rates while maintaining clean accuracy.

**Questions:**

What specific real-world scenario justifies the assumption that the defender only has access to a few clean samples?

Is the poisoning ratio consistent across all attacks, and if so, what value is used?

How does RVPT compare to standard fine-tuning on the same clean data in terms of both defense performance and parameter efficiency?

**Ethical Concerns:**

["NO or VERY MINOR ethics concerns only"]

**Final Justification:**

I appreciate the authors' thorough ablation studies and their effort to clarify aspects of the threat model through discussion. However, I remain concerned about several aspects that require further development before the work can be considered ready for publication.

Firstly, the threat model as currently presented appears overly limited and insufficiently realistic. The most critical assumption is related to the fact that the threat model assumes the defender has access to only a limited number of clear examples, without a clear justification for this constraint. However, in realistic scenarios, defenders often acquire models from sources where information about the training data is at least partially available, or where the underlying dataset is drawn from well-known public corpora. Alternatively, even when the training data is not publicly released (as in the case of CLIP), neither the defender nor the attacker typically has direct access to it (unless the attacker is the original model provider, such as OpenAI for CLIP), making it infeasible to execute the attack. In essence, in both cases, the attacker and defender may share comparable levels of access to potential data sources, which invalidates the asymmetry in data access assumed by the threat model in this paper.

Ultimately, although it is a minor point in the final assessment, the rebuttal now presents numerous experiments, clarifications on the threat model, and additional discussions that should be integrated into the main paper, necessitating a major revision of the paper.
Therefore, in light of the above, I maintain a slightly negative score but remain open and encourage further discussion with the other reviewers.

**Limitations:**

Yes! However, the limitations are discussed in the appendix but not in the main body of the paper. I encourage the authors to move this discussion into the main paper.

**Paper Formatting Concerns:**

No, the paper reflects the guidelines from NeurIPS 2025.

**Quality:**

2

**Strengths And Weaknesses:**

The paper is generally well-written, and the authors conduct multiple ablation studies to deepen the understanding of the proposed approach. The main idea is simple and shows promising results against multiple defenses.
However, there are several important points that warrant further consideration.

**Threat Model Clarity and Realism** The paper assumes a defender who has access to the backdoored model and a limited set of clean downstream data. However, the rationale behind this "few-shot clean data" constraint remains underspecified. In real-world settings, defenders often obtain models from repositories like HuggingFace, which typically include model cards and documentation about the training data, often enabling access to the same or a similar dataset. Suppose the assumption is instead that the model is trained privately and poisoned before deployment. In that case, the clean data likely originates from the same source, raising the question: why is only a small sample retained?

**Comparison to Simple Baselines.** RVPT is presented as a lightweight and highly effective defense strategy. However, the empirical evaluation does not convincingly isolate the contribution of its core components, nor does it compare to critical baseline alternatives. The most immediate missing baseline is standard fine-tuning (considering the same set of parameters utilized by RVPT) of the visual encoder on the few-shot clean data. This would establish whether RVPT provides a tangible advantage over conventional tuning strategies. Similarly, input preprocessing-based defenses are notably absent. Techniques such as JPEG compression [i] or test-time sanitization [ii-v] are well-known strategies that aim to erase or obfuscate triggers at inference time. Their inclusion would help gauge whether RVPT adds substantive value over these established techniques. Finally, the comparison lacks recent VLM-specific defenses such as RoCLIP [vi], which are particularly relevant given the paper's multimodal focus and claims regarding robustness on contrastive architectures.
To summarize, actionable points for the authors are to add comparisons to (i) fine-tuning on clean samples with the same parameter budget; (ii) at least a representative input-sanitization technique; and (iii) state-of-the-art methods like RoCLIP. Alternatively, clarify why certain baselines were excluded if they are not adopted.

[i] Xue et al. Compression-resistant backdoor attack against deep neural networks, 2023.

[ii] Liu et al. Neural trojans. 2017

[iii] Li et al. Backdoor attack in the physical world. 2021

[iv] Sun et al. Mask and restore: Blind backdoor defense at test time with masked autoencoder. 2023.

[v] Shi et al. Black-box backdoor defense via zero-shot image purification. 2023.

[vi] Yang et al. Robust contrastive language-image pretraining against data poisoning and backdoor attacks. 2024.

**Evaluation Coverage: Attacks and Poisoning Budget.** The set of attacks evaluated, while reasonably diverse, omits several stealthier or defense-aware backdoor methods. In particular, attacks like Reflection Backdoor [vii] and LIRA [viii] are known for being more covert and resilient to representation-based defenses. Given that RVPT relies on perturbation sensitivity, it may be less effective against these stronger attack strategies.

**Evaluation coverage on poisoning budget.** It is also unclear whether the poisoning budget is kept constant across all attacks, or what specific poisoning ratio is used. This aspect of the setup is left unspecified (or hard to find). More importantly, the paper lacks an ablation study exploring how RVPT performs as the poisoning ratio increases.

**Abstract figure**. The methodology section (3.3) introduces the main methodology proposed in this paper, which can be somewhat complex to understand from the text alone. Figure 2 attempts to provide a schematic overview but lacks an intuitive flow or entry point. Please consider this as minor feedback, which, however, could make the paper more accessible and reader-friendly.

**Related work.** The related work section primarily discusses classic defenses and some recent multimodal ones (CleanCLIP), but omits broader classes of defenses, especially input sanitization and prompt-based robustness as [i-vi].

**Terminology**. While the paper is generally well-written, a few choices in phrasing and terminology could be refined: The use of "perturbation resistivity" as a term is unusual and little related to its mathematical definition. From my perspective, a more direct term could be "embedding robustness". That said, this is a stylistic suggestion and does not affect the paper's overall quality.

The paper presents an interesting and promising approach. However, its contribution would benefit from a more precise presentation and formalization of the threat model, more comprehensive comparisons against state-of-the-art attacks and defenses, and a stronger connection to existing literature. The "few-shot clean data" assumption, as currently framed, lacks grounding in realistic scenarios. Additionally, the limited baseline scope makes it challenging to evaluate the practical applicability of the method in broader settings.

---

> ### Author Rebuttal · Authors · 2025-07-31
>
> Thank you for your insightful comments!
>
> ## W1: Unrealistic threat model
> Thanks for raising this concern. We agree that defenders may have access to the original training data for open-sourced pre-trained models and private training. However, there are also many scenarios where the **pre-training data of closed-source foundation models (such as OpenAI's CLIP or ChatGPT) is unavailable** to the defenders. In this case, defenders can only defend the potential backdoored model with their limited downstream data. Our assumption is specifically based on the scenario where defenders have no access to the pre-training data and can only use their own limited downstream data, which we believe is realistic since there are more and more closed-sourced models being released and adopted.
>
>
> ## W2: Lack of baseline comparison
> Thanks for raising this concern.
> **Comparison to standard fine-tuning.** We appreciate you highlighting the importance of including a standard fine-tuning baseline. This comparison is indeed presented in Figure 3(f) of our paper, where the result for standard fine-tuning (also referred to as VPT) corresponds to the case when the margin is set to 1. To further demonstrate RVPT's superiority in providing robustness against backdoor attacks, we also include comparisons with other parameter-efficient fine-tuning methods that share similar parameter budgets in Table 1.
>
> **Table 1：Attack Success Rate (Clean Accuracy) of multiple methods against different backdoor attacks. VPT represents standard fine-tuning with the same parameter budget as RVPT.**
> | Method | Random | Blended | BadCLIP |
> | :--- | :--- | :--- | :--- |
> | **CoOp** [1] | 45.65 (65.78) | 46.11 (65.77) | 99.73 (64.49) |
> | **Linear Probe** | 3.05 (59.64) | 5.52 (59.69) | 99.70 (59.33) |
> | **MaPLe** [2]| 61.72 (63.77) | 60.19 (63.92) | 98.85 (62.67) |
> | **VPT** [3] **(standard fine-tuning)**| 65.06 (40.20) | 64.93 (47.95) | 98.14 (64.32) |
> | **RVPT** | **0.05** (62.76)| **0.02** (62.36) | **2.76** (61.81) |
>
>
>
> **Comparison to other defense strategies.** Regarding other types of defenses, such as input preprocessing (e.g., JPEG compression) and in-training defenses (e.g., RoCLIP). We categorize these as orthogonal to our post-training approach. Our primary focus is on defending a model after it has been trained, ensuring it is not susceptible to manipulation by poisoned samples during inference. Differently, input-preprocessing defenses are test-time techniques that operate without modifying the underlying model, which remains backdoored. In-training defenses aim to prevent backdoors during the training phase, and thus assume access to and control over the training pipeline. Nevertheless, to further highlight RVPT's effectiveness, we compare RVPT directly with these defense baselines. Actually, since these defenses operate at different stages of the model lifecycle, RVPT can even be **combined** with them to achieve enhanced backdoor robustness. The results are summarized in Table 2.
>
> **Table 2: The comparisons of defense strategies with and without RVPT against BadCLIP.**
> |Method| JPEG-compression [4] | ZIP [5]|RoCLIP [6]|RVPT|
> | -------- |  -------- |-------- |--|--|
> |without RVPT|94.12 (55.88)|98.2 (61.40)|3.92 (61.53)| **2.76** (61.81) |
> |with RVPT| **0.14** (56.02) | **1.62** (62.31) |**0** (63.14)  | / |
>
>
> ## W3: Evaluation of more attacks
>
> Thanks for the suggestion. In response, we have expanded our evaluation to include the two recommended more covert backdoor attacks: LiRA and Reflection Backdoor. The results shown in Table 3, demonstrate that our method remains highly effective even against these stronger and more stealthy attack strategies.
>
> **Table 3: The performance of RVPT and CleanCLIP against two covert attacks.**
> | Method | Reflection [7]| LiRA [8]|
> | -------- | -------- | -------- |
> |No defense|  90.85 (63.06)    |  98.72 (63.15)    |
> |CleanCLIP| 17.99 (56.44)     |   14.25 (57.18)   |
> |RVPT|   **0** (63.58)   |  **0** (63.91)    |
>
> Although these attacks use triggers that are stealthy in the model's representation space, RVPT effectively neutralizes them. This robustness stems from RVPT's core mechanism: it identifies and prunes features from the pre-trained model that are non-essential to the downstream task. Consequently, if trigger patterns do not align with downstream task semantics, they are eliminated—regardless of how inconspicuous they may be in the representation space.
>
>
> ## W4: Evaluation of more poisoning budget
> Thanks for your insightful comments.
>
> **Poisoning budget settings.** We would like to clarify that the details of the poisoning budget are provided in Section 4.1 (Experiment Settings), specifically in **Lines 183-186**, where we outline the specific poisoning budget used for the attacks.
>
> **Performance across different poisoning ratios.** We fully agree that a robust defense should maintain effectiveness under varying levels of poisoning. To this end, Table 4 demonstrates that RVPT consistently performs well across a wide range of poisoning ratios.
>
> **Table 4: Performance of RVPT across different poisoning rates (pr). The default setting in the main paper corresponds to pr = 3e-3.**
> | Method | pr=1e-4 | pr=5e-4 |pr=1e-3 | pr=3e-3 |pr=1e-2 |  pr=1e-1 |
> | -------- | -------- |--- |-------- |-------- |-|-------- |
> |No defense|  0.03 (62.76) | 63.93 (62.87)  |  86.13 (62.85)    |99.83 (61.33)|   99.81 (62.98)   |   99.99 (62.83)  |
> |RVPT|   **0.02** (62.63)   | **0.09** (62.62)  |**1.09** (62.66)   | **2.76** (61.81)|  **2.07** (62.69)    |  **1.03** (62.76)   |
>
> ## Other minor issues
>
> We sincerely thank you for these constructive suggestions to improve the paper's clarity. In the revised manuscript, we will:
>
> - Redesign Figure 2 to have a more intuitive, step-by-step flow.
> - Expand the Related Work section to include more defenses, such as input-sanitization and prompt-based defenses.
> - Consider adopting your suggestion to use the term "embedding robustness" throughout the paper for better clarity.
>
> ## References
> [1] Learning to Prompt for Vision-Language Models, IJCV 2022.
>
> [2] MaPLe: Multi-modal Prompt Learning, CVPR 2023.
>
> [3] Visual prompt tuning, ECCV 2022.
>
> [4] Compression-resistant backdoor attack against deep neural networks, Applied Intelligence 2023.
>
> [5] Black-box Backdoor Defense via Zero-shot Image Purification, NeurIPS 2023.
>
> [6] Robust contrastive language-image pre-training against data poisoning and backdoor attacks, NeurIPS 2023.
>
> [7] Reflection backdoor: A natural backdoor attack on deep neural networks, ECCV 2020.
>
> [8] LIRA: Learnable, Imperceptible and Robust Backdoor Attacks, ICCV 2021.

---

> > ### Comment · Reviewer_g6BJ · 2025-08-04
> >
> > Thank you for the clarification.
> >
> > I appreciate that the authors explicitly situate their threat model in the context of closed-source foundation models where the original training data may not be accessible to the defender. However, I would argue that this scenario remains relatively narrow and context-specific.
> > Even when the training data is not publicly released (as in the case of CLIP or ChatGPT), it is often known (and declared) to be scraped from common, publicly accessible sources (e.g., LAION, Wikipedia). In such cases, both the attacker and defender may share comparable levels of access to potential data sources, invalidating the *asymmetry* threat model assumed between the attacker and defender in terms of data visibility.
> > Moreover, it is unclear under what concrete downstream application scenarios a defender would be limited to *only* a few clean examples while having full access to the model parameters and complete control over the recovery process. Could the authors elaborate on where this type of setup is encountered in practice? For instance, are there enterprise use cases or third-party auditing scenarios where such a data-limited but model-accessible defense is likely to appear?
> >
> >
> > Regarding the poisoning budget, I still find the experimental setup around poisoning ratios somewhat underspecified.
> > First, regarding the results reported in Table 4, it has not been explicitly stated which specific backdoor attacks these results correspond to. Are the numbers averaged across all attacks, or do they refer to a specific instance of an attack? Given that different attacks vary significantly in their stealth and effectiveness, a per-attack breakdown would help clarify whether RVPT's performance generalizes robustly across all trigger types.
> > Second, it is surprising to observe that the attack success rate decreases as the poisoning ratio increases, e.g., from 2.76% (pr = 3e-3) to 1.03% (pr = 1e-1). Intuitively, one would expect stronger poisoning to make the defense more challenging. Could the authors comment on why the defense appears more effective under higher poisoning rates? Is this effect consistent across all attacks, or does it reflect specific interactions between RVPT and certain triggers?
> > Finally, while the poisoning ratio used in the main experiments (e.g., 0.3% of the dataset) may reflect a plausible low-resource attacker scenario, it also represents a relatively limited and optimistic setting for evaluating a defense. In practice, a robust defense mechanism should be tested under a range of threat intensities and larger budgets (e.g., 5% or 10%, commonly used in the backdoor literature), including more challenging or worst-case scenarios.

---

> > > ### Author Response · Authors · 2025-08-05
> > > **W1: Clarification of the threat model**
> > >
> > > We really appreciate your valuable response to our rebuttal.
> > >
> > > ## W1: Clarification of the threat model
> > >
> > > ### W1.1: Availability of the pre-training data to the defenders
> > > It is true that there are many open-source models that fully or partially reveal their training data. However, we believe that studying the backdoor robustness of data-closed-source models remains crucial, as:
> > >
> > > 1. **The backdoor robustness of many multimodal models depends on CLIP, whose training data is entirely unavailable.** OpenAI does not reveal CLIP's pre-training data at all. They only stated that they searched for image–text pairs using a pre-set list of 500,000 queries, without specifying which public sources were used [1]. Since CLIP has become the backbone of many prevailing and widely used multimodal models [2,3,4] or applications, their vulnerability to manipulation by triggered input is directly related to the backdoor robustness of CLIP. Notably, recent work [5] shows that a backdoored CLIP can transfer its backdoor vulnerability to multimodal models where it serves as the visual encoder. Thus, although data-closed-source models may be fewer, their backdoor risks can propagate broadly to many downstream multimodal models. Defending backdoored CLIP is therefore critical, even when its training data is unavailable. Indeed, most prior works [6,7,8,9] on post-hoc defenses for backdoored multimodal models also adopt the same assumption that training data is not available to defenders.
> > >
> > > 2. **Even though the training data is available, leveraging it to defend multimodal pre-trained models is very challenging.** Since multimodal models are trained on image-text pairs, their training datasets are usually sourced from the Internet (many public image-text datasets [11,12,13] are also sourced from the Internet), which can be poisoned by the adversary in a very practical way [13]. Moreover, unlike supervised models, multimodal models are often trained on a larger dataset and can be compromised with significantly fewer poisoned examples [6]. Therefore, even if the training data is available, detecting poisoning samples in pre-training data and retraining or unlearning the poisoned samples is highly challenging and computationally prohibitive. An alternative is to use a surrogate image-text dataset that is declared or likely included in the pre-training (e.g., CC3M) for post-hoc defense. However, this is also suboptimal: RVPT instead leverages a novel loss that identifies predictive features specific to the downstream task and prunes non-predictive ones, yielding stronger robustness. In contrast, surrogate datasets suffer from mismatches in predictive features, reducing backdoor defense effectiveness even with FR loss of RVPT. Moreover, naive fine-tuning of backdoored models or other proposed post-training methods (CleanCLIP [7]) with such a surrogate dataset also results in suboptimal performance. Our experiments (Table 1) confirm these observations.
> > >
> > > **Table 1: ASR (CA) Performance against BadCLIP of different fine-tuning methods with or without FR loss.**
> > > |Fine-tuning method|without FR Loss| with FR Loss|
> > > |--|--|--|
> > > |Fine-tuning on CC3M using info-NCE Loss|92.50 (54.50)|18.23 (54.66)|
> > > |CleanCLIP|89.70 (57.55)|/|
> > > |RVPT on few-shot ImageNet|98.14 (64.32)|**2.76** (61.81) \[**our RVPT**\]|
> > >
> > > 3. **Even if leveraging pre-training data can be useful, it remains suboptimal compared to RVPT.** Such approaches require adding an additional cleansing stage into the pre-training or fine-tuning pipeline, which is time-consuming and difficult to deploy in practice. Specifically, CLIP must first be cleaned through post-training or retraining methods and only then adapted to the downstream task. In contrast, our RVPT provides end-to-end defense during adaptation, offering both higher effectiveness and greater efficiency.

---

> > > ### Author Response · Authors · 2025-08-05
> > > **W1: Clarification of the threat model (cont)**
> > >
> > > ### W1.2: Practical scenarios of the data-limited but model-accessible defense
> > > Due to being trained on large-scale image–text pairs, multimodal pre-trained models (e.g., CLIP) not only demonstrate remarkable zero-shot transferability but also strong few-shot adaptation ability [14]. Therefore, **there are many scenarios where practitioners leverage CLIP’s few-shot ability to adapt it into their tasks using limited data**, such as image recognition [14,15,16], dense prediction [17], and action recognition [18]. Notably, leveraging few-shot data to fine-tune CLIP is both popular and practical, as it enables enterprises with limited data (e.g., scenarios where labeled data is hard to collect) or limited computing resources to quickly incorporate CLIP into their pipelines.
> > >
> > > For instance, in the task of *medical imaging for rare diseases*, due to privacy and scarcity, the images of multiple rare diseases are difficult to collect. In these scenarios, a healthcare-providing enterprise may adapt CLIP with only a few-shot data. However, simple adaptation alone cannot guarantee backdoor robustness, leaving the model vulnerable to trigger-induced misdiagnoses that threaten patient safety and expose the provider to severe liability. The traditional approach uses image-text pairs that are likely to be in CLIP's pre-training dataset, which appears time-consuming and ineffective, while our RVPT can achieve **end-to-end defense**, ensuring backdoor robustness and adaptation to downstream tasks at the same time.
> > >
> > > To sum up, although CLIP can be quickly adapted to such downstream tasks, its training data collection process makes it possible to be backdoored or manipulated during deployment, raising significant security concerns. To address this limitation, we propose a very effective and efficient defense of RVPT, leveraging the limited available data **not only to adapt CLIP to downstream tasks, but also to safeguard its backdoor robustness at the same time.**

---

> > > ### Author Response · Authors · 2025-08-05
> > > **W2: More results regarding the poisoning ratio**
> > >
> > > ## W2: More results regarding the poisoning ratio
> > >
> > > ### W2.1: RVPT's generalization to all trigger types across all poisoning ratios
> > > First, we apologize for the earlier ambiguity. The trigger type in our previous experiments corresponds to BadCLIP, one of the state-of-the-art multimodal backdoor attacks.
> > >
> > > To demonstrate that the effectiveness of RVPT generalizes across both different poisoning ratios and different triggers, we now include additional results for BadNet (patch-based trigger), Blended (global-based trigger), WaNet (human-imperceptible trigger), and ISSBA (instance-dependent trigger) in Table 1, demonstrating RVPT's defending ability can generalize well to all poisoning ratios and all trigger types.
> > >
> > > **Table 1: ASR (CA) of RVPT under different poisoning ratios with different triggers.**
> > > | Method | pr=1e-4 | pr=5e-4 |pr=1e-3 | pr=3e-3 |pr=1e-2 |  pr=1e-1 |
> > > | -| -|-|-|-|-|-|
> > > |BadCLIP – No def.|0.03 (62.76)| 63.93 (62.87)|86.13 (62.85)|99.83 (61.33)|99.81 (62.98)|99.99 (62.83)|
> > > |BadCLIP – RVPT|**0.02** (62.63)|**0.09** (62.62)|**1.09** (62.66)|**2.76** (61.81)|**2.07** (62.69)|**1.03** (62.76)|
> > > | BadNet – No def.|0 (63.52)|0.06 (63.53) | 17.78 (63.59) |82.69 (63.04) | 89.97 (63.32) | 95.79 (62.47) |
> > > | BadNet – RVPT| **0** (63.16) | **0** (63.21) |**0.01** (63.73)|**0.05** (62.76)|**0.10** (63.05) | **0.14** (63.88)|
> > > | Blended – No def.| 0.02 (63.34) | 39.91 (63.17) | 84.30 (63.17) |98.52 (62.64) |99.83 (63.05) | 99.98 (62.82) |
> > > | Blended – RVPT| **0** (63.58) | **0.02** (63.57) | **0.01** (62.98) | **0.02** (62.36) | **0.02** (63.68) | **0** (63.73) |
> > > | WaNet – No def.| 0.15 (62.73) | 60.90 (62.85) | 78.50 (62.80) | 87.18 (62.42) | 88.05 (62.95) | 92.47 (62.12) |
> > > |WaNet – RVPT| **0.04** (62.61) | **0.16** (62.60) | **1.35** (62.67) | **0.03** (62.48) | **2.00** (62.70) | **0** (62.96) |
> > > |ISSBA – No def.| 0.05 (62.74) | 40.20 (62.83) | 55.60 (62.80) |60.01 (61.72) | 61.30 (62.93) | 63.51 (62.82) |
> > > |ISSBA – RVPT| **0.03** (62.62) | **0.15** (62.61) | **1.10** (62.66) | **0.01** (61.92) | **1.80** (62.69) | **0.90** (62.75) |
> > >
> > > ### W2.2: Why is RVPT more effective under higher poisoning ratios?
> > >
> > > We believe that this reflects a trade-off between **attack effectiveness** and the **entanglement of clean and backdoor features.** As a recent work [19] shows, a low poisoning ratio (pr) tends to cause such entanglement, which undermines the effectiveness of tuning-based defenses. Since RVPT aims to eliminate irrelevant features, it is also slightly affected under the low pr scenario, making it perform better at the high pr scenario. However, at a low pr, the attack itself is less effective, as the model may fail to memorize the spurious relationship between trigger and target class due to the limited number of poisoned samples.
> > >
> > > Thus, RVPT’s performance across pr is best understood as a balance between reduced attack strength (at low ratios) and increased feature entanglement, rather than a simple monotonic relationship where higher poisoning always reduces defense effectiveness. As a result, shown in Table 1, there are different trends in different triggers.
> > >
> > >
> > > ### W2.3: RVPT's effectiveness at a high poisoning ratio of 10%
> > > We acknowledge that a robust defense should be evaluated in a large poisoning ratio, like 10%. The reason why we set the poisoning rate to 0.3% in our main experiment is that a recent paper [6] shows multimodal pre-trained models can already be effectively backdoored with as little as 0.01% of the training samples, which is several orders of magnitude lower than what is typically required for supervised models. Thus, we believe using pr of 0.3% is sufficient to backdoor a multimodal model. Consistently, most prior works on multimodal backdoor attack [6,20] and defense [7,8,9] adopt the same poisoning rate.
> > >
> > > Further, to address your concern, we also report results under a 10% ratio in Table 2, confirming that RVPT remains effective in more challenging scenarios.
> > >
> > > **Table 2: ASR (CA) of RVPT under the poisoning rate of 10% with different triggers.**
> > > | Method | BadNet | Blended |ISSBA | WaNet |TrojVQA | BadCLIP |
> > > |-|-|-|-|-|-|-|
> > > |No defense|  95.79 (62.47) | 99.98 (62.82)  |  63.51 (62.85)|92.47 (62.12)|   99.79 (63.01) |   99.99 (62.83)|
> > > |RVPT|**0.14** (63.88)|**0** (63.73)|**0.90** (62.75)|**0** (62.96)|**0** (63.69)|**1.03** (62.76)|

---

> > > ### Author Response · Authors · 2025-08-05
> > > **References**
> > >
> > > ## References
> > > [1] Learning transferable visual models from natural language supervision, ICML 2021
> > >
> > > [2] Visual instruction tuning, NeurIPS 2024
> > >
> > > [3] Improved baselines with visual instruction tuning, CVPR 2024
> > >
> > > [4] BLIP-2: Bootstrapping Language-Image Pre-training with Frozen Image Encoders and Large Language Models, ICML 2023
> > >
> > > [5] Stealthy Backdoor Attack in Self-Supervised Learning Vision Encoders for Large Vision Language Models, CVPR 2025
> > >
> > > [6] Poisoning and Backdooring Contrastive Learning, ICLR 2022
> > >
> > > [7] CleanCLIP: Mitigating Data Poisoning Attacks in Multimodal Contrastive Learning, ICCV 2023
> > >
> > > [8] Unlearning Backdoor Threats: Enhancing Backdoor Defense in Multimodal Contrastive Learning via Local Token Unlearning, CVPR 2024
> > >
> > > [9] A Closer Look at Backdoor Attacks on CLIP, ICML 2025
> > >
> > > [10] CoCa: Contrastive Captioners are Image-Text Foundation Models, arXiv 2022
> > >
> > > [11] A cleaned, hypernymed, image alt-text dataset for automatic image captioning, ACL 2018
> > >
> > > [12] LAION-5B: An open large-scale dataset for training next generation image-text models, arXiv 2022
> > >
> > > [13] Poisoning Web-Scale Training Datasets is Practical, S&P 2024
> > >
> > > [14] Learning to Prompt for Continual Learning, IJCV 2022
> > >
> > > [15] Conditional Prompt Learning for Vision-Language Models, CVPR 2022
> > >
> > > [16] MaPLe: Multimodal Prompt Learning, CVPR 2023
> > >
> > > [17] DenseCLIP: Language-Guided Dense Prediction with Context-Aware Prompting, CVPR 2022
> > >
> > > [18] ActionCLIP: A new paradigm for video action recognition, arXiv 2021
> > >
> > > [19] Towards Stable Backdoor Purification through Feature Shift Tuning, NeurIPS 2024
> > >
> > > [20] BadCLIP: Dual-Embedding Guided Backdoor Attack on Multimodal Contrastive Learning, CVPR 2024

---

> > > > ### Comment · Reviewer_g6BJ · 2025-08-05
> > > >
> > > > Dear authors, thank you once again for your responses. At this stage, I believe the clarifications provided are satisfactory. I will take some time to further reflect on the discussion and will consult with the other reviewers to reach a final evaluation.

---

> > > > > ### Author Response · Authors · 2025-08-06
> > > > >
> > > > > Dear Reveiwer g6BJ，
> > > > >
> > > > > Thank you very much for your valuable contribution to the rebuttal. We are very delighted to see that our responses are satisfactory to you. If you still have any concerns about our work, we are more than willing to engage in further discussion with you.
> > > > >
> > > > > Best regards,
> > > > >
> > > > > Authors

---

> ### Author Response · Authors · 2025-08-04
> **Looking forward to further feedback**
>
> Dear Reviewer g6BJ,
>
> Thank you again for your helpful feedback! We would be grateful if you could let us know whether our response has sufficiently addressed your concerns. If there are any remaining issues or points that need further clarification, we would be glad to continue the discussion.
>
> Best regards,
>
> The Authors of Paper 19327

---

### Note · Authors · 2025-08-12

We thank the AC and reviewers for their constructive feedback. We are pleased that all four reviewers acknowledged the value of our work and found our rebuttal clarifying.

- **Threat-model realism.** Our setting targets **data-closed multimodal models** for three main reasons:

   1. **CLIP is data-closed and widely reused.** As a core backbone in many downstream applications and prevailing multimodal models, any backdoor in CLIP can propagate to these dependent systems. Therefore, strengthening the backdoor robustness of CLIP—a data-closed model—should be a priority.
   2. **Cleansing via pre-training data is highly challenging even when it is available.** Image–text corpora used for multimodal pre-training are (a) easily poisoned, and (b) highly vulnerable: a few poisoned pairs among millions of clean pairs can suffice to implant a backdoor. Detecting such poisons and retraining at this scale is both computationally prohibitive and operationally challenging.
   3. **Surrogate data approaches are suboptimal.** Some methods assume access to a surrogate dataset that might overlap with the pre-training data. However, such datasets suffer from predictive-feature mismatch with the downstream task, leading to weaker defenses, and they require multi-stage pipelines, making them less efficient and deployable than our end-to-end RVPT.

- **Baselines and coverage.** We added (i) standard fine-tuning with the same parameter budget (VPT), (ii) parameter-efficient methods (CoOp, MaPLe), and (iii) input-preprocessing (JPEG, ZIP) and in-training defenses (RoCLIP). RVPT consistently outperforms these or complements them when combined.

- **Evaluation breadth.** We expanded to stealthier attacks (Reflection, LIRA), cross-dataset/domain and emerging-target settings, and poisoning ratios from 1e-4 to 1e-1. RVPT remains robust across triggers and threat intensities; higher poisoning can even make triggers more separable, improving FR pruning effectiveness.

- **Key outcome.** RVPT tunes only 0.27% of parameters, reduces ASR to <3% while maintaining clean accuracy, generalizes across datasets/domains/target classes, and can integrate with orthogonal defenses for further gains.

We believe these clarifications directly address the reviewer’s concerns regarding realism, baselines, and evaluation coverage, and further demonstrate RVPT’s practicality and robustness.

---

### Decision · Program_Chairs · 2025-09-17

**Decision:**

Accept (poster)

**Comment:**

This paper proposes a new defense method against backdoors on CLIP. All four reviewers have actively engaged in the discussion and appreciate the technical novelty and extensive experiments (in both the original paper and multi-round responses). Although Reviewer g6BJ is still concerned about the realism of the threat model (assuming that the defender only has access to a few clean samples), they clearly stated in the final justification and the confidential comment to AC that this concern is just a minor one by saying "Overall, I am not strongly opposed to this paper, so I would not recommend a clear rejection." and "Ultimately, although it is a minor point in the final assessment...". The AC also checked the authors' response to Reviewer g6BJ and noticed that the authors have already provided new results to show the proposed method is still better than using a substitute dataset of clean samples (see Table 1: ASR (CA) Performance against BadCLIP of different fine-tuning methods with or without FR loss).

Overall, the AC thinks that the strengths of this paper slightly outweigh its limitations and therefore recommends acceptance. However, this paper still requires a major revision before publication by incorporating all the new results and clarifications. The final decision would be given based on the quality of the revised paper.